# Transcription attenuation amplifies collateral vulnerabilities in rifampicin-resistant *Mycobacterium tuberculosis*

Kathryn A. Eckartt[1,3], Vanisha Munsamy-Govender [1,3], Stefany Quiñones-Garcia[1], Michael A. DeJesus [1], Xiangwu Ju [2], Shixin Liu [2] & Jeremy M. Rock [1] ✉

*Mycobacterium tuberculosis* (Mtb) acquires resistance to rifampicin (Rif) through mutations in the β-subunit of RNA polymerase (RNAP) that prevent the drug from binding. The most common mutation is a single amino acid substitution, βS450L, that confers antibiotic resistance. This mutation also results in collateral effects that impact bacterial physiology and fitness, although the mechanisms underlying many of these effects remain unclear. Here we employed a CRISPRi comparative functional genomics approach to analyse gene vulnerability differences between βS450L Mtb and two alternative Rif-resistant (RifR) Mtb strains, βD435V and βH445Y. Among the strongest βS450L-specific vulnerabilities, we identified thiamine and branched-chain amino acid (BCAA) biosynthesis pathways. These vulnerabilities arise, at least in part, due to transcription attenuation, which impairs βS450L Mtb's ability to upregulate expression of the critical BCAA biosynthetic enzyme *ilvB1* in response to genetic or chemical inhibition. Together, our findings highlight the distinct physiological impacts of RifR in Mtb, identify transcription attenuation as a key driver of βS450L-specific vulnerabilities, and suggest potential avenues for targeted intervention.

Antimicrobial resistance poses a growing challenge to public health. *Mycobacterium tuberculosis* (Mtb), the causative agent of tuberculosis (TB), exemplifies this crisis: drug-resistant Mtb accounts for one-fifth of all deaths due to antimicrobial resistance and Mtb is the single leading cause of death due to infectious disease[1]. While drug-resistance mutations enhance bacterial survival in the presence of cognate antibiotics, they can also alter bacterial physiology and reduce fitness when antibiotics are absent[2-10]. These physiological changes can render the bacteria more or less susceptible to the inhibition of other pathways. Such so-called collateral phenotypes form the basis of rational drug combinations designed to slow the evolution of resistance[11-15]. However, the underlying mechanisms driving these collateral effects, particularly in Mtb, remain incompletely understood.

Rifampicin (Rif) is a potent first-line antibiotic and a cornerstone of modern TB treatment. Its introduction effectively halved the duration of TB therapy[16]. However, decades of Rif use have led to the emergence of rifampicin-resistant (RifR) Mtb, which now contributes to hundreds of thousands of TB cases[1]. Rif exerts its antibacterial effect by binding to the β-subunit of RNA polymerase (RNAP) and blocking the extension of short RNA transcripts[17]. Mtb acquires resistance to Rif through mutations in the β-subunit that prevent the drug from binding, with the most common mutation (~70% of all RifR Mtb)[6] being a single amino acid substitution−serine 450 to leucine (βS450L)[6]. RifR mutations, including βS450L, alter the shape and chemical properties of the Rif-binding pocket, which is involved in forming the elongating RNA exit pathway[17]. As a result, RifR mutations have the potential to impact the RNA elongation step of transcription[18]. Potentially consistent with this hypothesis, we and others have shown that the fitness cost associated with the βS450L mutation is at least partly due to increased transcriptional pausing and termination by βS450L RNAP[18,19].

[1]Laboratory of Host-Pathogen Biology, The Rockefeller University, New York, NY, USA. [2]Laboratory of Nanoscale Biophysics and Biochemistry, The Rockefeller University, New York, NY, USA. [3]These authors contributed equally: Kathryn A. Eckartt, Vanisha Munsamy-Govender. ✉e-mail: rock@rockefeller.edu

Beyond conferring antibiotic resistance, the βS450L mutation affects the sensitivity of several cellular processes to genetic inhibition, a phenomenon we refer to as differential vulnerabilities[19]. Studying these differential vulnerabilities provides insight into the physiological consequences of RifR and may also help identify targeted therapeutic strategies that exploit emergent weaknesses in drug-resistant strains.

Here we employed a comparative functional genomics approach to investigate the mechanistic basis of differential vulnerabilities in βS450L. Using our tunable CRISPR interference (CRISPRi) system, we assessed gene vulnerability across three RifR Mtb strains: βS450L, βD435V and βH445Y, each encoding a RifR RNAP variant with distinct kinetic properties[18,20,21]. Our analysis revealed that βS450L-specific vulnerabilities include thiamine and branched-chain amino acid (BCAA) biosynthesis. We further demonstrate that these vulnerabilities are driven, at least in part, by transcription attenuation, which limits the ability of βS450L Mtb to upregulate expression of key biosynthetic genes in response to pathway inhibition. This attenuation defect appears to be a broader consequence of the βS450L mutation, driving increased vulnerability in numerous pathways. Collectively, our findings highlight the distinct physiological consequences of the three most common RifR variants in Mtb, establish attenuation as a driver of βS450L-specific vulnerabilities, and suggest potential avenues for targeted therapeutic interventions aimed at exploiting collateral vulnerabilities in RifR Mtb.

## Results

### A genetic screen to identify differential vulnerabilities in 'fast' RifR RNAP mutants

The fitness cost of the βS450L mutation is at least partly due to increased RNAP pausing and termination[19]. Previously, we identified approximately 150 genes involved in diverse processes that show altered sensitivity to genetic silencing in βS450L Mtb (hereafter, βS450L), which we refer to as differential vulnerabilities[19]. These differential vulnerabilities could represent primary or secondary consequences of altered transcription dynamics in βS450L. Alternatively, they might represent non-specific effects, such as altered genetic requirements due to the slower growth rate of βS450L or altered RNAP stability[22,23].

To prioritize hit genes linked to the altered transcription dynamics of βS450L, we used a comparative functional genomics approach, analysing differential vulnerabilities in two additional RifR mutants, βD435V and βH445Y (Fig. 1a,b). These mutations represent the second and third most common RifR variants in clinical Mtb strains[6]. Similar to βS450L, both impose a fitness cost in the absence of rifampicin, although the βD435V cost is media dependent (Extended Data Fig. 1a,b)[5]. However, unlike βS450L, which encodes a slow, pause-prone RNAP, βD435V and βH445Y encode fast, pause-resistant RNAPs[18,21].

To identify differential vulnerabilities in βD435V and βH445Y Mtb, we screened these strains using our tunable CRISPRi library (Fig. 1b), which contains 96,700 sgRNAs targeting ~98% of annotated Mtb genes[24,25]. For biosafety reasons, screens were performed in isogenic RifR biotin auxotrophs (ΔbioA), which grow normally with 2 μM biotin but cannot establish infection in mice[26]. Unless otherwise noted, follow-up experiments were performed in prototrophic H37Rv Mtb. Rifampicin-sensitive (RifS) and βS450L ΔbioA Mtb screens were conducted previously[19].

After transforming the CRISPRi library into βD435V and βH445Y ΔbioA Mtb, we performed two independent competitive growth experiments. Triplicate cultures were propagated for approximately 30 generations with or without the CRISPRi inducer anhydrotetracycline (ATc; Fig. 1b). Genomic DNA was collected every 2.5 or 5 generations, followed by deep sequencing to assess sgRNA abundance between induced and uninduced conditions. Growth phenotypes were well correlated across replicates (Extended Data Fig. 1c–f). Whole-genome sequencing of the final competitive growth timepoint confirmed that reversion to RifS rpoB alleles did not take over the culture at late timepoints.

sgRNA depletion data were used to determine gene vulnerability using a multilevel Bayesian model, as previously described[19,24]. Rather than making binary gene essentiality calls, this model generates expression–fitness relationships for each targeted gene by relating the predicted magnitude of target knockdown (as inferred by the sgRNA strength) to the resulting fitness cost imposed on the bacteria. The model iterates to generate distributions of vulnerability values for each gene. Differential vulnerabilities were defined as genes whose 95% credible interval for the vulnerability difference excluded zero.

To assess screen quality, we first compared differential gene vulnerability between RifS, βD435V and βH445Y strains. As expected, most genes showed similar vulnerability across all three strains ($R^2 = 0.955$ and $R^2 = 0.938$; Fig. 1c,d and Supplementary Table 1), as illustrated by the nicotinamide adenine dinucleotide (NAD) biosynthetic enzyme nadB (Fig. 1e). However, 43 genes in βD435V and 137 genes in βH445Y exhibited significant changes in vulnerability (|Δvulnerability| ≥ 3) relative to RifS (Fig. 1c–i and Supplementary Table 1). Genes whose inhibition imposes a greater fitness cost in the RifR strain than in RifS are classified as collateral vulnerabilities, whereas genes whose inhibition leads to a smaller fitness cost in RifR are classified as collateral invulnerabilities. In βD435V, the numbers of genes in each category were roughly balanced, whereas βH445Y showed a pronounced skew towards collateral invulnerabilities. This asymmetry may reflect distinct effects of the βD435V and βH445Y mutations on RNAP structure and the transcription cycle, or from other uncharacterized mutation-specific differences, despite both being considered 'fast', pause-resistant RNAP mutants[18,21]. Notably, βD435V and βH445Y did not share any strong collateral vulnerabilities, although they displayed substantially more overlap in collateral invulnerabilities (Supplementary Table 1).

### Identification of βS450L-specific differential vulnerabilities

To identify genes specifically vulnerable in βS450L, we compared gene vulnerability between βS450L, βD435V and βH445Y. As expected, most genes showed similar vulnerability between strains (Extended Data Fig. 2a–c). Strikingly, however, most genes identified as collateral vulnerabilities in βS450L were either not hits or were called as collateral invulnerabilities in βD435V and βH445Y (Fig. 2a, Supplementary Table 1 and Extended Data Fig. 2a–c). Further comparisons between βS450L and βD435V or βH445Y revealed significant

**Fig. 1 | A genetic screen to identify differential vulnerabilities in 'fast' RifR RNAP mutants. a**, Structural model of an Mtb RNAP transcription initiation complex bound to Rif. Inset: location of the three RifR mutations analysed in this paper. **b**, Quantification of differential vulnerabilities (ΔV). An anhydrotetracycline (ATc)-inducible CRISPRi library was transformed into βD435V or βH445Y ΔbioA Mtb. RifS and βS450L screens were performed previously[19]. Genes essential for in vitro growth were targeted with sgRNAs of varying predicted knockdown efficiencies. (**i**) Cultures were passaged and sgRNA abundance was assessed by deep sequencing at multiple timepoints. (**ii**) A Bayesian model was then applied to estimate the expression–fitness relationship (curved lines) for each gene. In brief, the x axis represents predicted sgRNA strength, which serves as a proxy for the extent of target gene knockdown, while the y axis shows bacterial fitness, measured as the log₂-transformed fold change (L2FC) of sgRNA abundance after 25 generations in the competitive growth experiment. Gene vulnerability was quantified by calculating the area above the expression–fitness curve. A gene was considered differentially vulnerable if the 95% credible region of its vulnerability difference did not overlap with zero. **c,d**, Scatterplot showing gene vulnerability (circles) in RifS and βD435V or βH445Y Mtb. Genes with the strongest differential vulnerabilities (|ΔV| ≥ 3) are highlighted in blue. **e–i**, Expression–fitness relationships for an example non-hit gene (nadB) and differential vulnerabilities (dnaJ2, gltB, rv1708 and ppk1). The light-coloured lines represent fits generated from 1,000 samples drawn from the posterior distributions, while the dark lines indicate the mean fit.

shifts in gene-level vulnerabilities across these RifR strains (Fig. 2b,c), indicating that although all three mutations confer RifR, their genetic vulnerabilities differ markedly.

## Thiamin biosynthesis is a βS450L-specific collateral vulnerability

One of the most differentially vulnerable genes in βS450L relative to βD435V and βH445Y was *thiS* (*rv0416*), which encodes a protein required for thiamine biosynthesis (Fig. 2b,c). While *thiS* is a strong collateral vulnerability in βS450L, it is a collateral invulnerability in βD435V and βH445Y (Fig. 3a). Thiamine (vitamin B1) is the precursor for thiamin pyrophosphate (TPP), an essential coenzyme[27]. Thiamine biosynthesis is essential for Mtb growth in vitro and in mice, and has been proposed as a potential source of drug targets[24,28].

We confirmed that *thiS* is a collateral vulnerability in βS450L. While targeting *thiS* with moderate-strength sgRNAs had little effect in wild

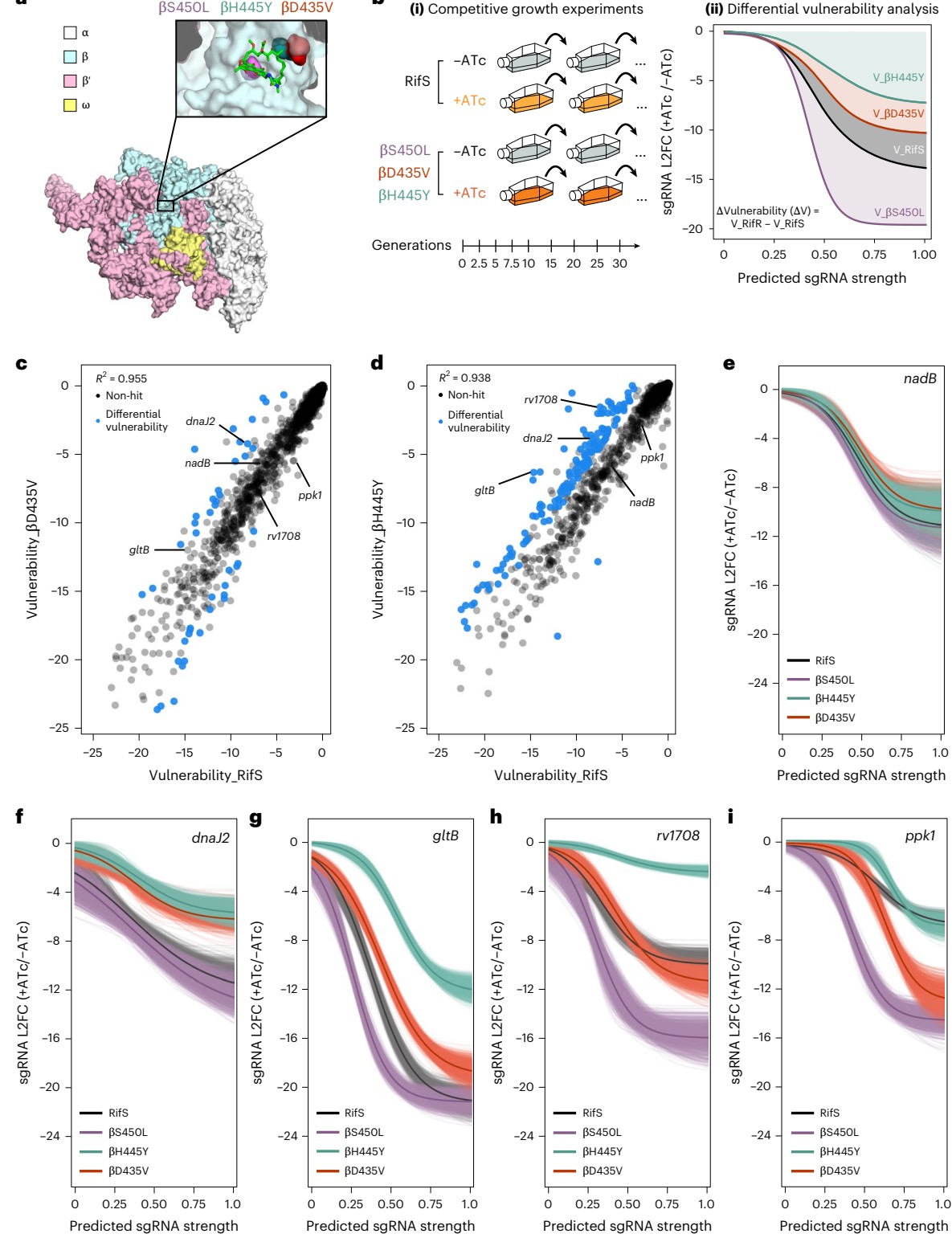

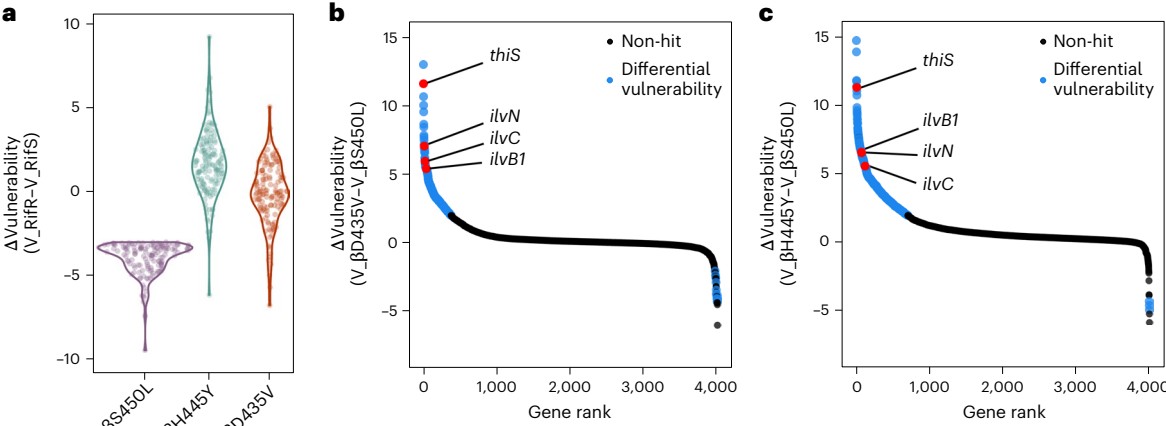

**Fig. 2 | Identification of βS450L-specific differential vulnerabilities.**
**a**, Comparison of collateral vulnerabilities associated with the βS450L mutation in the three RifR mutants. Each circle represents an individual gene classified as a collateral vulnerability in the βS450L mutant (95% credible region of its vulnerability difference did not overlap with zero and ΔV ≤ −3). The y axis represents ΔV, defined as the difference in gene vulnerability values between a RifR strain and the RifS strain. **b,c**, Differential vulnerabilities in βD435V and βH445Y compared to βS450L RifR strains. Ranked scatterplots depict the ΔV values for all genes, ranked by ΔV on the x axis. Each dot corresponds to a single gene targeted in the library. Black dots are non-hit genes, blue dots show significant differences in vulnerability (|ΔV| ≥ 2), and red dots highlight specific hit genes.

type (WT), βD435V and βH445Y, it significantly inhibited the growth of βS450L (Fig. 3b). In contrast, targeting the control gene *nadB* produced similar fitness costs across strains (Fig. 3b). Thiamin is derivatized in a multistep pathway to TPP. Many of the enzymes involved in converting thiamin to TPP are also collateral vulnerabilities in βS450L (Supplementary Table 1). Collectively, these findings confirm that TPP synthesis is a collateral vulnerability specific to βS450L.

Mtb encodes 12 enzymes known or predicted to require TPP for activity[29] (Fig. 3c). These enzymes participate in various cellular processes, including glycolysis (pyruvate dehydrogenase, *aceE*), the TCA cycle (alpha-ketoglutarate dehydrogenase, *kgd*), branched-chain amino acid biosynthesis (acetohydroxyacid synthase, *ilvB1*) and other metabolic pathways. Among these 12 TPP-dependent enzymes, two were strong collateral vulnerabilities in βS450L. The weaker hit, *dxs1* (*rv2682c*), both utilizes TPP and is essential for TPP biosynthesis (Fig. 3c)[30]. The stronger hit, *ilvB1* (*rv3003c*), encodes an essential enzyme that catalyses the first common step in branched-chain amino acid biosynthesis (Fig. 3c)[31,32]. These findings suggest that the increased vulnerability of TPP biosynthesis in βS450L is driven, at least in part, by a linked collateral vulnerability in *ilvB1*.

## BCAA biosynthesis is a βS450L-specific collateral vulnerability
IlvB1 is a biosynthetic enzyme essential for the de novo synthesis of BCAAs valine, leucine and isoleucine, as well as for the production of pantothenate, the precursor to coenzyme A[32]. IlvB1 is required for Mtb growth in vitro and in mice[32].

*ilvB1* is part of a three-gene operon with *ilvN*, encoding the regulatory subunit for acetohydroxyacid synthase, and *ilvC*, another BCAA biosynthetic enzyme. Similar to *thiS*, *ilvB1*, *ilvN* and *ilvC* are among the most differentially vulnerable genes in βS450L compared to βD435V and βH445Y (Figs. 4a and 2b,c). Beyond *ilvB1*, inhibition of multiple enzymes involved in BCAA and pantothenate biosynthesis and utilization imposed a greater fitness cost in βS450L (Fig. 4b). While none of these fitness costs are as severe as that associated with *ilvB1*, this underscores the vulnerability of this metabolic pathway in βS450L.

We confirmed that *ilvB1* is a collateral vulnerability in βS450L. Whereas targeting *ilvB1* with strong sgRNAs blocked growth of all strains, a moderate-strength sgRNA only imposed a fitness cost in βS450L (Fig. 4c). Importantly, this fitness cost was rescued by BCAA and pantothenate supplementation. Moreover, consistent with the hypothesis that the collateral vulnerability of *thiS* is at least in part

due to the dependence of IlvB1 activity on TPP, BCAA and pantothenate supplementation also partially restored growth in βS450L *thiS* knockdown strains (Fig. 3d).

Because of its essentiality in Mtb, the absence of a human homologue and the successful targeting of IlvB1 homologues in other species, Mtb IlvB1 has attracted considerable attention as a drug target[31,33]. To determine whether genetic vulnerability of IlvB1 in βS450L translates to increased chemical vulnerability, we tested chlorflavonin, a fungal secondary metabolite known to inhibit Mtb IlvB1 (ref. 33).

To validate its on-target activity, we analysed chlorflavonin using our CRISPRi chemical-genetics platform[34]. This approach leverages a genome-scale CRISPRi library to modulate Mtb gene expression and measure bacterial fitness in the presence of a growth-inhibitory compound, identifying genes whose inhibition alters bacterial fitness under compound exposure. In some cases, a compound's direct target emerges as a highly sensitizing hit, as CRISPRi-mediated knockdown reduces the inhibitor dose needed to suppress bacterial growth[34]. Consistent with published on-target activity[33], *ilvB1* was one of the top sensitizing hits in both RifS and βS450L Mtb (Fig. 4d, Extended Data Fig. 3a and Supplementary Table 2). *thiS* and other thiamin biosynthetic genes were also top sensitizing hits, confirming the importance of TPP for IlvB1 activity (Fig. 4d and Supplementary Table 2). To test the sensitivity of βS450L to chlorflavonin, we performed minimum inhibitory concentration (MIC) assays. As predicted by the genetic screen, βS450L was more sensitive to chlorflavonin than RifS, βD435V or βH445Y (Fig. 4e). Importantly, this phenotype was not limited to the reference Mtb strain H37Rv; βS450L was also more sensitive to chlorflavonin than RifS or βH445Y in the lineage 2 Mtb strain HN878 (Extended Data Fig. 3b).

## Elevated transcription attenuation drives the *ilvB1* collateral vulnerability in βS450L
When analysing the *ilvB1NC* operon, we observed that it contains an unusually long leader sequence. In Mtb, the median leader length, excluding leaderless transcripts, is ~48 nucleotides, whereas the *ilvB1NC* leader spans 166 nucleotides[35,36]. This long leader suggested a potential regulatory function.

To examine *ilvB1NC* expression, we profiled RifS Mtb using simultaneous end sequencing (SEnd-seq), which maps the 5' and 3' ends of individual RNA molecules[36]. This revealed an abundant ~120-nucleotide transcript that terminates within the leader sequence, upstream of the *ilvB1* open reading frame (ORF) (Fig. 5a and Supplementary Table 3),

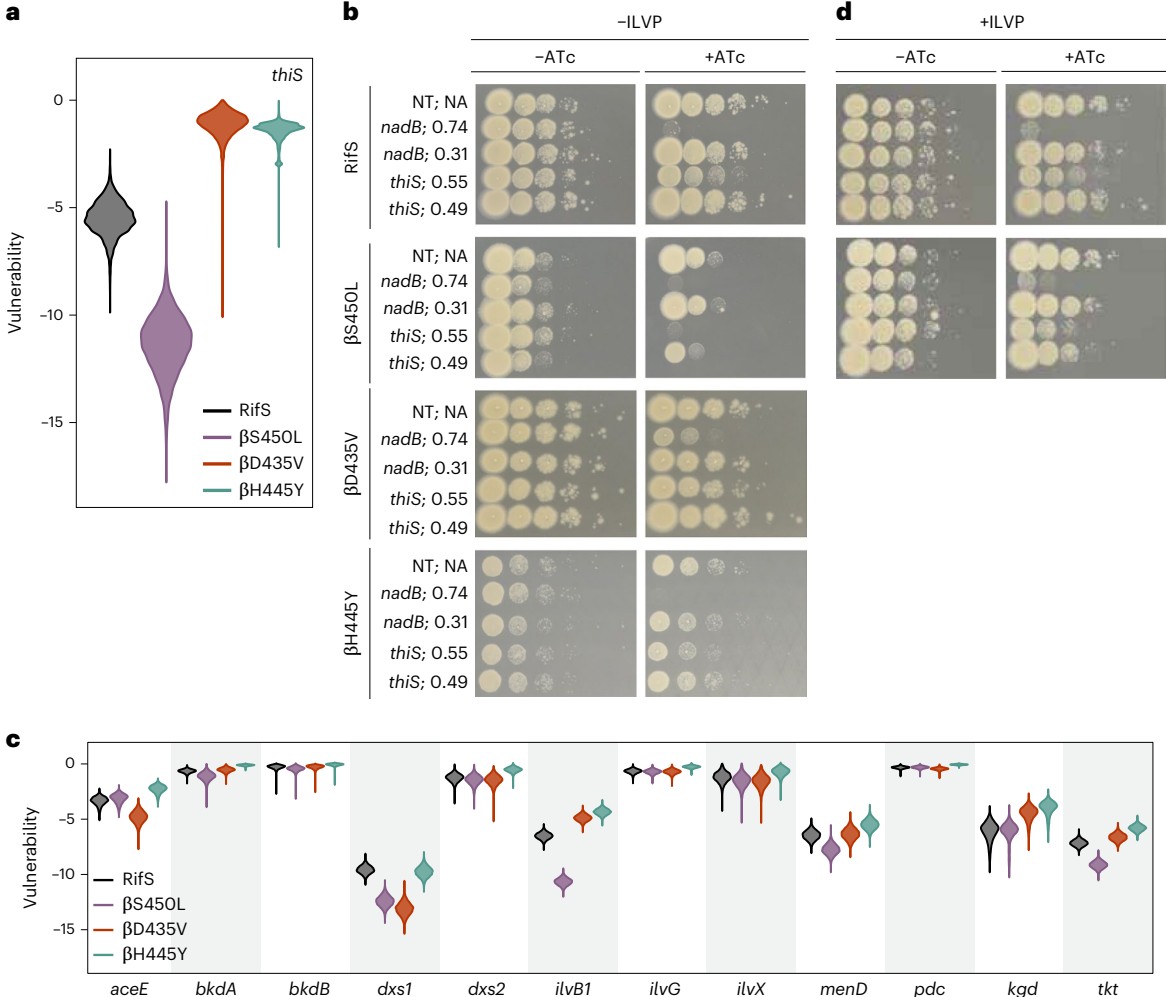

**Fig. 3 | Thiamin biosynthesis is a βS450L-specific collateral vulnerability.**
**a**, Vulnerability distributions for *thiS* in the indicated Mtb strains. **b**, Phenotypic consequences of knockdown of *nadB* and *thiS* in RifS and RifR Mtb. sgRNA-predicted strength (a proxy for the magnitude of target knockdown) is shown next to each strain, where 1.0 represents the strongest possible sgRNA.

NT, non-targeting sgRNA. **c**, Vulnerability distributions for the indicated TPP-dependent gene products in the indicated Mtb strains. **d**, Phenotypic consequences of knockdown of *nadB* and *thiS* in RifS and βS450L Mtb as in **b**, except that plates are supplemented with 50 μg ml⁻¹ isoleucine, leucine, valine and pantothenate (ILVP).

consistent with previous observations[35]. Further examination of this transcript identified a short upstream ORF (uORF) enriched in BCAA, which appears to be actively translated based on published ribosome footprinting data[37] and *lacZ* fusion experiments[35] (Fig. 5a). These leader sequence features share key characteristics of a regulatory mechanism called transcription attenuation, in which the expression of biosynthetic enzymes is modulated by the availability of their end products[38]. On the basis of these observations, we hypothesized that, similar to *ilvB1* regulation in *E. coli* and other bacteria[39], the *ilvB1NC* operon in Mtb may also be controlled by transcription attenuation.

The proposed transcription attenuation mechanism for the *ilvB1NC* operon in Mtb operates as follows (Fig. 5b,c). The leader sequence can fold into one of two mutually exclusive RNA secondary structures: a terminator or an anti-terminator. As RNAP transcribes the leader sequence, the ribosome initiates translation of the uORF. Within the uORF, there are four regulatory codons encoding 'LVVI', which function as sensors for the availability of aminoacylated transfer RNAs specific to BCAA. When BCAAs are abundant, the ribosome translates the uORF without stalling and reaches the stop codon. In doing so, it masks ~26 nucleotides of the messenger (m)RNA[37], preventing formation of the anti-terminator structure. This allows the downstream Rho-independent terminator[38] to form, terminating transcription

and suppressing *ilvB1NC* expression. In contrast, when BCAAs are scarce, the ribosome stalls at the 'LVVI' regulatory codons within the uORF, allowing the anti-terminator to form. This prevents terminator formation, allowing RNAP to continue transcribing the *ilvB1NC* coding regions. As a result, key enzymes involved in BCAA biosynthesis are produced, helping to restore BCAA levels.

This attenuation mechanism provides a plausible explanation for why *ilvB1* is a collateral vulnerability in βS450L. Unlike βD435V and βH445Y, βS450L encodes a slow RNAP that exhibits increased transcription pausing and termination[18,20,21]. We hypothesized that the collateral vulnerability of *ilvB1* in βS450L is driven, at least in part, by elevated transcription termination within the *ilvB1* leader. As a result, βS450L has a reduced ability to upregulate *ilvB1* expression in response to *ilvB1* inhibition.

Supporting this hypothesis, published quantitative proteomics show that IlvB1 is modestly but significantly underexpressed (~44% lower than wild type) in a βS450L isolate of Mtb CDC1551 grown under standard laboratory conditions[40]. When this βS450L strain acquired a compensatory *rpoC* mutation (Leu516Pro), IlvB1 protein levels returned to wild type[40]. *rpoC* compensatory mutations have previously been shown to partially alleviate the elevated transcription termination defect of βS450L RNAP[18,41].

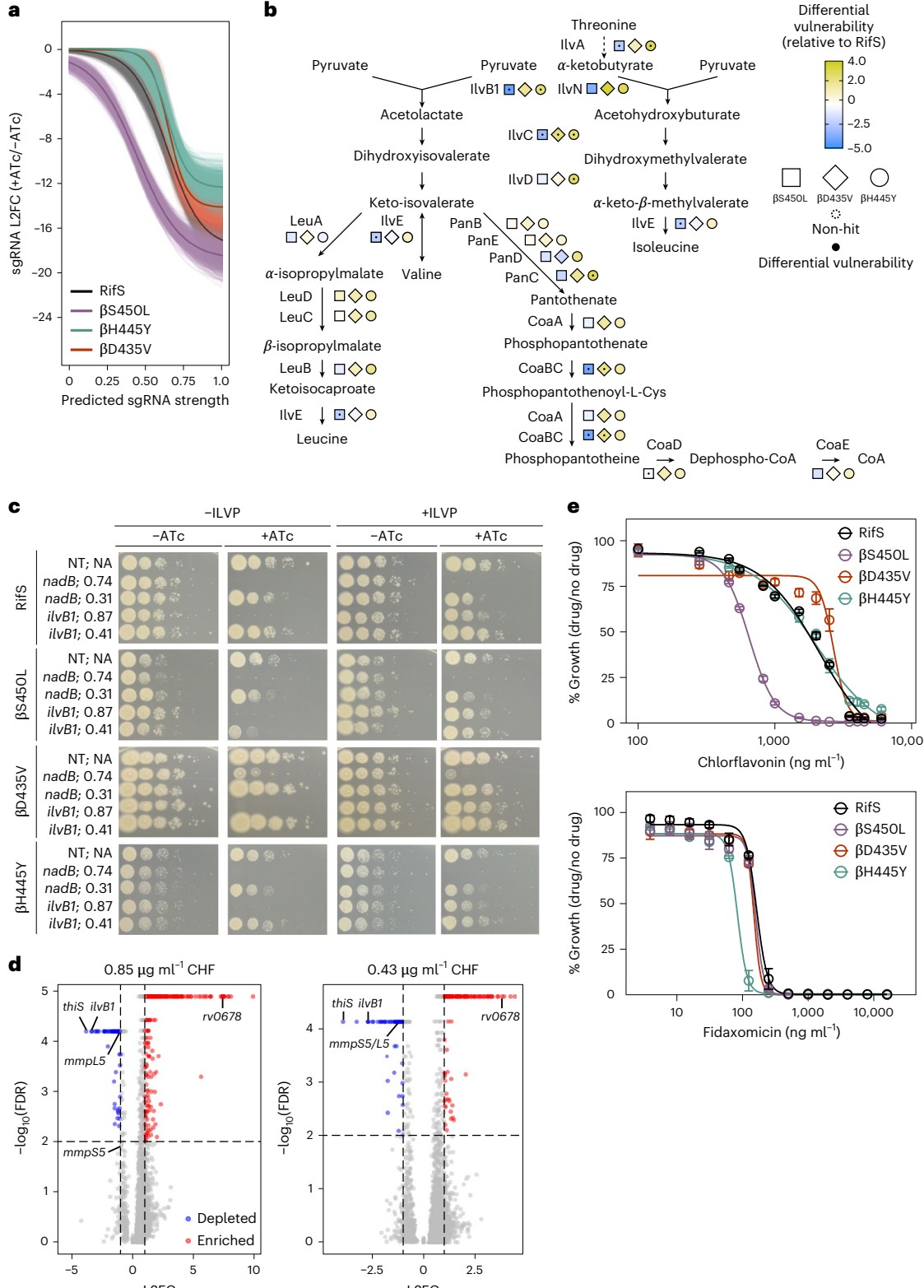

**Fig. 4 | BCAA biosynthesis is a βS450L-specific collateral vulnerability.**
**a**, Expression–fitness relationships for *ilvB1* in the indicated Mtb strains. The light-coloured lines represent fits generated from 1,000 samples drawn from the posterior distributions, while the dark lines indicate the mean fit. **b**, Differential vulnerability values (RifR–RifS) for genes involved in the biosynthesis of BCAA, pantothenate and coenzyme A (CoA). Blue or yellow colouring indicates that the specified gene is either more or less vulnerable, respectively, in the corresponding RifR strain. A black dot indicates that the differential vulnerability is statistically significant (95% credible interval for the difference in vulnerability did not include

zero). Squares, βS450L; diamonds, βD435V; circles, βH445Y. **c**, Phenotypic consequences of knockdown of *nadB* and *ilvB1* in RifS and RifR Mtb as in Fig. 3b,d. **d**, Volcano plots showing L2FC values and false discovery rates (FDR) for each gene (circles) in the presence of two sub-MIC doses of chlorflavonin (CHF). Negative L2FC values indicate that knocking down the target gene sensitizes Mtb to CHF, while positive values indicate increased resistance. Significantly depleting or enriching hits (FDR ≤ 0.05 and |L2FC| ≥ 2 as determined by MAGeCK linear modelling[71]) are shown in blue and red, respectively. **e**, Dose–response curves (mean ± s.e.m., *n* = 3 biological replicates) for the indicated strains.

The attenuation model predicts that *ilvB1NC* expression should increase in response to IlvB1 inhibition and a subsequent decrease in BCAA levels. Supporting this prediction, RNA-seq and quantitative (q)PCR confirmed strong *ilvB1NC* upregulation following chlorflavonin treatment (Fig. 5d, Extended Data Fig. 3c,d and Supplementary Table 4). To directly test whether βS450L has a reduced ability to upregulate *ilvB1* expression following IlvB1 inhibition, we designed fluorescent reporter plasmids[42] driven by either a strong constitutive promoter (*P300-mScarlet*) or the *ilvB1* promoter and leader sequence (*PilvB1-mScarlet*). As predicted, βS450L was less effective at increasing *ilvB1* reporter expression in response to chlorflavonin treatment, while expression from the control P300 promoter remained unchanged (Fig. 5e and Extended Data Fig. 3e,f). In contrast, βD435V and βH445Y showed higher baseline *PilvB1-mScarlet* expression and stronger induction, consistent with the hypo-terminating properties of these RNAP (Fig. 5e)[18,20,21]. Further supporting the proposed regulatory mechanism, mutating the putative 'LVVI' regulatory codons to 'AAAA' eliminated chlorflavonin-dependent upregulation (Fig. 5f and Extended Data Fig. 3g), confirming the role of these codons in transcription attenuation.

To test the prediction that βS450L terminates more efficiently in the *ilvB1* leader, we designed in vivo reporter constructs to measure termination efficiency. We used a constitutive promoter to drive the expression of a bicistronic mRNA encoding two reporter proteins, separated by either a terminator sequence or a control non-terminator sequence. βS450L displayed higher termination levels at the *rrf* terminator in vivo, consistent with previous in vitro findings (Fig. 5g)[19]. Similarly, βS450L also showed increased termination at the *ilvB1* terminator (Fig. 5g), supporting the hypothesis that excessive termination drives, at least in part, the collateral vulnerability of *ilvB1* in βS450L.

If transcription attenuation underlies this vulnerability, suppressing over-termination should mitigate the collateral phenotype. Consistent with this, βS450L carrying the compensatory *rpoC* F452L mutation, which partially suppresses the over-termination phenotype of βS450L[18], displayed intermediate sensitivity to chlorflavonin (Fig. 5h). Moreover, disruption of the Rho-independent terminator within the *ilvB1* attenuator similarly mitigated the collateral vulnerability phenotype (Fig. 5i). Together, these findings provide strong evidence that enhanced transcription attenuation is a key driver of the *ilvB1* collateral vulnerability in βS450L Mtb.

## Discussion

In previous work, we identified differential vulnerabilities in βS450L Mtb affecting diverse cellular processes[19]. Here we investigate the mechanisms underlying these collateral effects. By expanding our CRISPRi comparative functional genomics approach to include two additional RifR mutants, βD435V and βH445Y, we identified βS450L-specific vulnerabilities. Among these, *thiS* and thiamin biosynthesis emerged as critical for βS450L fitness due to a linked collateral vulnerability in the thiamin-dependent enzyme IlvB1, which functions in BCAA and pantothenate biosynthesis. We further showed that the IlvB1-related vulnerability is driven, at least in part, by transcription attenuation that limits βS450L's ability to upregulate IlvB1 following inhibition. Collectively, these findings reveal physiological consequences of RifR in Mtb and suggest potential therapeutic opportunities.

Biochemical studies show that βS450L RNAP elongates more slowly and terminates more frequently than RifS RNAP[18,21], consistent with recent in vivo observations[19]. We propose that these kinetic differences impair βS450L's ability to upregulate *ilvB1* by reducing readthrough of the *ilvB1* attenuator. Transcription attenuation regulates *ilvB1* in other bacteria[39], although its occurrence in Mtb was previously uncertain[43]. Analysis of the *ilvB1* leader reveals a conserved BCAA-rich uORF across mycobacterial species (Extended Data Fig. 4). Our hypothesis is further supported by foundational work from Horn and Yanofsky showing that RifR mutants selected for altered transcription of the *E. coli trp* attenuator—including βH445Y—affect termination efficiency[20]. The molecular details of transcription attenuation in Mtb remain underexplored, making it difficult to determine exactly why βS450L exhibits increased termination at the *ilvB1* attenuator. In principle, two simple mechanisms could explain this effect: (1) increased intrinsic termination at the terminator or (2) reduced formation of the anti-terminator.

We propose that attenuation defects extend beyond *ilvB1* and represent a broader regulatory consequence of the βS450L mutation.

**Fig. 5 | Elevated transcription attenuation drives, at least in part, the collateral vulnerability of *ilvB1* in βS450L Mtb. a,b**, SEnd-seq data track showing total RNA coverage of the *ilvB1* 5′ leader region. +1 marks the transcription start site (TSS), +126 indicates the end of the U-tract, and +168 indicates the start of the structural gene sequences. Ribosome profiling[72] demonstrates ribosome occupancy within the *ilvB1* uORF. Depicted are ribosome footprinting read counts overlayed on the *ilvB1* 5′ leader and 5′ end of the *ilvB1* structural gene. **c**, Predicted RNA folding of the anti-terminator and terminator hairpins in the *ilvB1* 5′ leader region. The ribosome footprint is estimated as a 26-nucleotide occlusion centred on the bolded regulatory (GUA) or stop codon (UGA)[37]. The hairpin minimum free energy ($\Delta G_{min}$) was calculated using the putative exposed RNA sequences using RNAfold[73]. **d**, Differentially expressed genes when RifS Mtb is treated with chlorflavonin (CHF, 8 μg ml$^{-1}$) or the vehicle control. Genes with $P_{adj} < 0.05$ and $|\log_2 FC| > 2$, as determined by DESeq2 linear modelling[74], are shown in purple and green. **e**, Regulation of the *ilvB1* 5′ leader was monitored by fusing the *mScarlet* fluorescent protein to the *ilvB1* promoter and leader sequence (*PilvB1-mScarlet*). This reporter plasmid was then integrated into the chromosome of the four indicated Mtb strains. mScarlet fluorescence normalized to culture optical density (RFU/OD$_{700}$) was monitored as a function of increasing CHF doses. Ethambutol (EMB) served as a control drug (1 μg ml$^{-1}$ = 1× MIC$_{90}$ for RifS Mtb). '*' indicates a significant difference determined by multiple unpaired, two-sided Welch *t*-tests with multiple hypothesis correction (Benjamini, Krieger and Yekutieli, 5% FDR). 'NS' indicates a non-significant difference by the same metric. Adjusted *P* values comparing RifS to βS450L, βD435V and βH445Y for 0 μg ml$^{-1}$: $4.41 \times 10^{-4}$, $1.00 \times 10^{-5}$, $1.84 \times 10^{-4}$; for 0.175 μg ml$^{-1}$: 0.030, $2.22 \times 10^{-3}$, $1.28 \times 10^{-4}$; for 0.7 μg ml$^{-1}$: 0.016, $8.89 \times 10^{-3}$, 0.004585; for 1.01 μg ml$^{-1}$: 0.10, 0.027, 0.10; for 1.4 μg ml$^{-1}$: $2.41 \times 10^{-3}$, 0.018, 0.36; for 2.8 μg ml$^{-1}$: 0.085, 0.11, 0.085; for 5.05 μg ml$^{-1}$: 0.029, 0.78, 0.029.

**f**, The four putative BCAA regulatory codons 'LVVI' (see **c**) in the mScarlet reporter described in **e** (*PilvB1-mScarlet*) were mutated to 'AAAA' (*PilvB1-Ala-mScarlet*). The introduced mutations are not predicted to affect the folding of the anti-terminator or terminator hairpins. This reporter was then integrated into the chromosome of either RifS or βS450L Mtb. Reporter activity was quantified as in **e**. This reporter shows a low, CHF dose-independent expression of mScarlet in both RifS and βS450L Mtb ($q = 0.101$ by unpaired *t*-test). **g**, Termination efficiencies in Mtb quantified with a dual-reporter system. The strong Ptb38 promoter drives expression of the reporter genes *lacZ* and luciferase, which are separated by an intervening terminator of interest. The relative expression of luciferase (RLU) and LacZ (RFU) was then used to quantify relative terminator readthrough in different Mtb strains. Three reporters encoding either no terminator (None) or the *rrf* or *ilvB1* terminator were integrated into the chromosome of the four indicated Mtb strains. Bar plots depict the readthrough (that is, termination efficiency) for each reporter. '*' indicates a significant difference determined by multiple unpaired, two-sided Welch *t*-tests with multiple hypothesis correction (Benjamini, Krieger and Yekutieli, 5% FDR). From left to right: *q* values for the *rrf* terminator are 0.005, 0.01 and 0.44; *q* values for the *ilvB1* attenuator are $1.72 \times 10^{-4}$, $3.21 \times 10^{-4}$ and $2.30 \times 10^{-5}$. **h**, CHF dose–response curves for RifS (black), βS450L (purple) and two isolates of the compensated RifR strain βS450L β′F452L (medium and light blue) Mtb. **e–h**, Data points show the mean and s.d. of 3 technical replicates and are representative of 2 independent experiments. **i**, Phenotypic consequences of *ilvB1* knockdown and complementation in βS450L. Complementation was achieved by ectopic expression of the *ilvB1* operon from a chromosomally integrated plasmid, driven by either the wild-type (WT) *ilvB1* promoter and 5′ leader or a mutant 5′ leader lacking the Rho-independent terminator (RIT-less). Knockdown was mediated by the same hypomorphic *ilvB1*-targeting sgRNA (0.41) used in Fig. 4c.

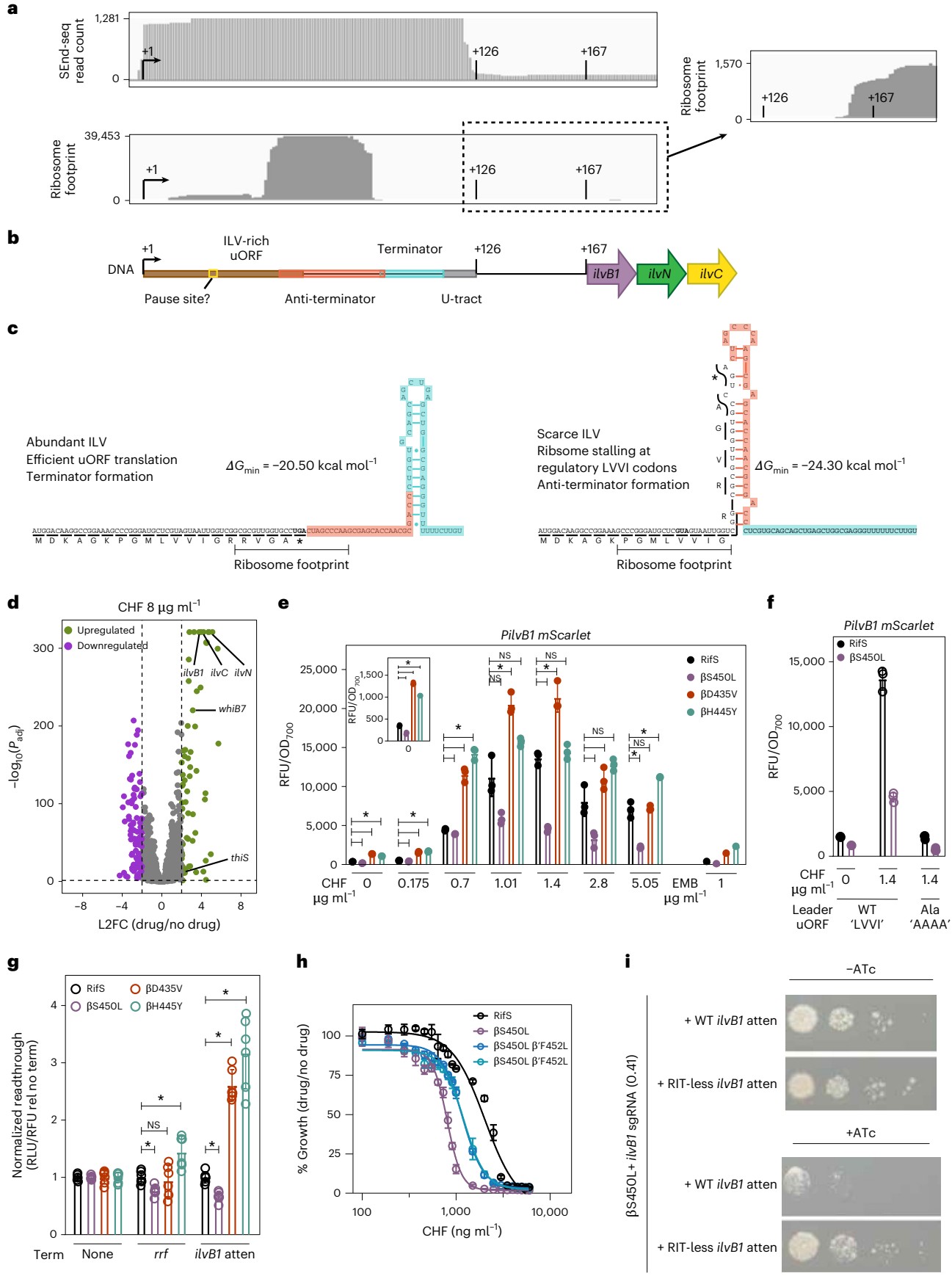

Because transcription attenuators are widespread in bacteria, elevated termination at these elements could contribute to multiple collateral vulnerabilities. Consistent with this idea, several predicted attenuation-regulated loci overlap with βS450L-specific vulnerabilities (Extended Data Fig. 5). One well-characterized example is the transcription factor *whiB7* (refs. 42,44,45) (Extended Data Fig. 6a). Ribosome stalling within the uORF induces *whiB7* expression, which then drives a transcriptional programme that includes intrinsic antibiotic resistance factors, the alanine biosynthesis gene *aspC* and other pathways. Similar to *ilvB1*, βS450L shows a defect in *whiB7* induction (Extended Data Fig. 6b,c), although we note that this impairment does not increase sensitivity to translation inhibitors under standard MIC conditions (Extended Data Fig. 6d,e). Nevertheless, just as the *thiS* collateral vulnerability is explained at least in part by a metabolically linked vulnerability in *ilvB1*, the vulnerability associated with *aspC*—a direct WhiB7-activated gene—may similarly reflect impaired *whiB7* activation (Extended Data Fig. 6f). Thus, attenuation defects in βS450L may propagate through regulatory circuits or metabolically linked pathways.

Beyond transcription, attenuation defects in βS450L may also affect translational control. In translation attenuation systems, base pairing between a uORF and downstream sequences can occlude the Shine–Dalgarno (SD) site and prevent translation initiation; ribosome stalling within the uORF disrupts this pairing, unmasking the SD and enabling translation of downstream genes[46]. Consistent with a broader role for attenuation defects in βS450L, several cysteine biosynthetic genes—recently shown to be regulated by translation attenuation in *M. smegmatis*[47]—are βS450L-specific collateral vulnerabilities, along with additional candidate loci (Extended Data Fig. 7). Although the molecular details remain unclear, these findings suggest that βS450L may disrupt attenuation-based regulation at both transcriptional and translational levels.

RifR mutations impose a fitness cost in Mtb[3–5], and compensatory mutations arise in *rpoC*, *rpoA*, *rpoB* and *nusG*[9,19,41,48,49]. If increased termination at the *ilvB1* attenuator reduces fitness in clinical βS450L isolates, compensatory mutations might arise within the attenuator itself. However, analysis of ~51,000 clinical Mtb genomes revealed no evidence for such mutations[19,34]. This is not entirely surprising, as compensatory mutations in RNAP or NusG can mitigate βS450L hyper-termination defects and broadly restore gene expression. We note, however, that rare mutations in the *whiB7* 5′ leader—known or predicted to decrease transcription termination—do occur and can promote drug resistance in Mtb[50]. It is conceivable that such variants could be selected in βS450L backgrounds. Selection might be driven by aminoglycosides such as streptomycin and amikacin, which are comparatively poor inducers of *whiB7* (ref. 42), under in vivo selective pressures not fully recapitulated by standard laboratory MIC testing conditions. Further work will be required to determine whether these variants arise preferentially in βS450L and whether they meaningfully compensate for attenuation-driven vulnerability in *whiB7* expression.

Although the fast RifR mutants were primarily used here as comparators to better understand βS450L, their vulnerability profiles are intriguing in their own right. Interestingly, despite both encoding fast, pause-resistant RNAPs[18,21], βD435V and βH445Y display distinct vulnerability profiles that may reflect yet-undefined mutation-specific consequences. βD435V showed a roughly balanced distribution of collateral vulnerabilities and invulnerabilities, whereas βH445Y was strongly skewed towards invulnerabilities. The basis for this difference remains unclear. One possibility is that βH445Y RNAP more effectively displaces dCas9 from DNA, reducing CRISPRi repression, although it is unclear why this would be specific to βH445Y and not shared with βD435V. A procedural difference between the screens is that βH445Y required a low rifampicin dose (0.05 μg ml⁻¹) to prevent reversion to wild type. Although this dose does not affect growth of βH445Y

Mtb[51] and is 1,000-fold below the half-maximal inhibitory concentration ($IC_{50}$) for inhibition of the equivalent RNAP mutant in *E. coli*[52], rifampicin could exert other subtle cellular effects. Finally, βH445Y alters cell envelope composition[53], which could modestly reduce ATc uptake and CRISPRi induction. Although the shifts in vulnerability were generally modest and not observed for all genes—and do not affect the conclusions regarding the specific hits examined here—this possibility represents a plausible contributing factor.

Collateral sensitivity is an evolutionary trade-off in which resistance to one antibiotic increases sensitivity to another[2,11–13,54]. The collateral vulnerabilities identified here may similarly represent collateral sensitivities[13], suggesting that inhibitors targeting these pathways could serve as companion drugs with rifampicin. Ideally, such targets would be shared across multiple RifR mutants. However, the distinct vulnerability patterns of βS450L, βD435V and βH445Y suggest that identifying a universal target may be difficult. Screens performed under conditions that better mimic the infection environment may reveal additional vulnerabilities, some of which may be conserved across mutants. Alternatively, allele-specific targeting may be advantageous. Because βS450L is the most prevalent clinical RifR mutation and the most fit variant in vitro[2–6,8,9], targeting this allele could shift evolutionary trajectories towards less-fit resistance variants. It is also important to note that CRISPRi mimics non-competitive inhibition, whereas small molecules can produce a wider range of biochemical effects; thus, CRISPRi screens capture only a subset of possible exploitable collateral sensitivities.

While our proof-of-principle data suggest that targeting IlvB1 could help slow the evolution of βS450L, its potential as a drug target presents both opportunities and challenges. On the positive side, Mtb IlvB1 has been validated as druggable, with multiple chemically distinct inhibitors that block Mtb growth in macrophages and mice[31,33,55,56]. However, despite strong attenuation of an *ilvB1* knockout in mice, the bacterium can persist for weeks after infection[32], possibly by scavenging BCAA from the host[57]. In addition, Mtb encodes several *ilvB1* paralogues (*ilvB2*, *ilvG* and *ilvX*), which could provide functional redundancy[58]. Further work is needed to define the contributions of IlvB1-dependent synthesis, paralogue redundancy and BCAA scavenging during infection. Resolving these dynamics will determine whether IlvB1 or other attenuation-linked collateral vulnerabilities (Extended Data Figs. 5–7) are viable companion targets for rifampicin.

Our findings highlight the complexity of allele-specific collateral effects in RifR. Different β-subunit mutations can have opposing effects on RNAP activity—for example, some slowing elongation while others accelerate it—leading to divergent collateral phenotypes. In contrast, resistance mutations in other targets may have more uniform effects on target activity and bacterial physiology, potentially producing more consistent collateral vulnerabilities suitable for rational combination therapy. Systematic studies of resistance mutations across diverse targets will be needed to test this idea.

Drug-resistant bacteria pose an escalating global threat[1]. Beyond conferring resistance, drug-resistance mutations can alter the biochemical properties of the encoded gene product, leading to changes in bacterial physiology and gene vulnerability. Our findings reveal distinct physiological consequences of RifR in Mtb, identify attenuation as a mechanism that increases gene vulnerability in βS450L, and suggest strategies for targeted intervention. More broadly, this approach can be applied to other pathogens to identify and exploit vulnerabilities associated with drug resistance.

## Methods

### Bacterial strains

Mtb strains are derivatives of H37Rv unless otherwise noted. *ΔbioA* Mtb was obtained from the Dirk Schnappinger laboratory[26]. *E. coli* strains are derivatives of DH5α (NEB). Strains are available upon reasonable request to the corresponding author.

## Mycobacterial cultures

Mtb were grown at 37 °C in Difco Middlebrook 7H9 broth or on 7H10 agar supplemented with 0.2% glycerol (7H9) or 0.5% glycerol (7H10), 0.05% Tween-80, 1× oleic acid-albumin-dextrose-catalase (OADC, Mtb) and the appropriate antibiotics (kanamycin 10–20 µg ml$^{-1}$ and/or hygromycin 25–50 µg ml$^{-1}$ and/or zeocin 5–20 µg ml$^{-1)}$). ATc was used at 100 ng ml$^{-1}$. Note that both 7H9 and 7H10 media are normally supplemented with biotin (0.5 mg l$^{-1}$, ~2 µM), thereby allowing growth of the Δ*bioA* Mtb auxotroph. Mtb cultures were grown standing in tissue culture flasks (unless otherwise indicated) with 5% $CO_2$. Relative growth of individual CRISPRi strains was determined by spotting assay. Tenfold serial dilutions (starting at 50,000 cells per spot) were plated on 7H10 with or without 100 ng ml$^{-1}$ ATc. Plates were incubated at 37 °C and imaged after 14 days.

## Selection of Rif-resistant isolates

For the selection of RifR H37Rv and Δ*bioA*, 5 independent 5-ml cultures were started at a density of ~2,000 cells per ml (to minimize the probability of seeding cultures with pre-existing RifR bacilli) and grown to stationary phase (optical density at 600 nm (OD$_{600}$) > 1.5). Cultures were pelleted (~3,900 × *g* for 10 min), resuspended in 30 µl remaining medium per pellet and plated on 7H10 agar supplemented with Rif at 0.5 µg ml$^{-1}$. After outgrowth, colonies were picked into 7H9 medium. After 1 week of outgrowth, an aliquot was heat inactivated and the Rif-resistance-determining region of *rpoB*, *rpoA* and *rpoC* were amplified by PCR and Sanger sequenced (see Supplementary Table 5).

## Generation of structural models

The structural model of Mtb RNAP transcription initiation complex bound to Rif was generated by modelling *Mycobacterium smegmatis* RNAP bound to Rif (PDB 6CCV)[59] onto the transcription initiation complex structure (PDB 6EDT)[60].

## Generation of individual CRISPRi strains

plRL58 (Addgene, 166886; Supplementary Table 5) contains (1) the *Sth1 dcas9* allele under the control of an optimized, synthetic Tet repressor (TetR)-regulated promoter; (2) the *Sth1* sgRNA under the control of a synthetic TetR-regulated promoter; (3) a mycobacterial codon optimized Tet repressor; (4) a single-copy L5-integrating backbone[61], with the integrase removed to increase plasmid stability; (5) a pBR322-derived *E. coli* replication origin; and (6) a kanamycin-selectable marker. To integrate plRL58 into the mycobacterial chromosome, L5 integrase function is supplied in *trans* on a separate suicide vector, plRL19 (Addgene, 163634; Supplementary Table 5).

Individual CRISPRi plasmids were cloned by restriction ligation cloning[62]. In brief, plRL58 was digested with BsmBI-v2 (NEB, R0739L) and gel purified. sgRNAs were designed to target the non-template strand of the target gene ORF. For each individual sgRNA, two complementary oligonucleotides with appropriate sticky end overhangs were annealed and ligated (T4 ligase, NEB, M0202M) into the BsmBI-digested plasmid backbone. Successful cloning was confirmed by Sanger sequencing. Individual CRISPRi plasmids were then electroporated into the appropriate mycobacterial species. Electrocompetent cells were obtained as previously described[63]. In brief, a bacterial culture was expanded to OD$_{600}$ = 0.4–0.6 and treated with glycine (final concentration 0.2 M) for ~1 doubling before pelleting (~3,900 x *g* for 10 min). The cell pellet was washed three times in sterile 10% glycerol. The washed bacilli were then resuspended in 10% glycerol in a final volume of 5% of the original culture volume. For each transformation, 100 ng plasmid DNA and 50–100 µl electrocompetent mycobacteria were mixed and transferred to a 2 mm electroporation cuvette (Bio-Rad, 1652082). Where necessary, 100 ng plasmid plRL19 (Addgene, plasmid 163634) was also added. Electroporation was performed using the Gene Pulser X cell electroporation system (Bio-Rad, 1652660) set at 2,500 V, 700 Ω and 25 µF. Electroporated cells were immediately recovered in 1 ml of

complete 7H9 and cultured at 37 °C for 1 doubling, after which cells were pelleted at ~1,800 × *g* for 10 min, resuspended in 100 µl of 7H9 and plated on selective 7H10 plates. Colonies were picked after 14–21 days.

## CRISPRi library transformation

Fifty transformations were performed to generate RifS, βS450L, βD435V and βH445Y Δ*bioA* libraries. For each transformation, 1 µg of RLC12 plasmid DNA was added to 100 µl (Δ*bioA*) electrocompetent cells. The cells:DNA mix was transferred to a 2 mm electroporation cuvette (Bio-Rad, 1652082) and electroporated at 2,500 kV, 700 Ω and 25 µF. Each transformation was recovered in 2 ml 7H9 medium supplemented with OADC, glycerol and Tween-80 (100 ml total) for 16–24 h. The recovered cells were collected at ~1,800× *g* for 10 min, resuspended in 400 µl remaining medium per transformation and plated on 7H10 agar supplemented with kanamycin (see 'Mycobacterial cultures') in Corning Bioassay dishes (Sigma, CLS431111-16EA). Note that the βH445Y library was generated and passaged in the presence of 0.05 µg ml$^{-1}$ Rif (2.5× RifS MIC) to select against bacilli that reverted to a WT *rpoB* allele.

After 21 days of outgrowth on plates, transformants were scraped into 25% glycerol and pooled. Scraped cells were homogenized by two dissociation cycles on a gentleMACS Octo Dissociator (Miltenyi Biotec, 130095937) using the RNA_01 programme and 30 gentleMACS M tubes (Miltenyi Biotec, 130093236). The Mtb libraries were further declumped by passaging 1–3 ml of homogenized library into 100 ml of 7H9 supplemented with kanamycin (see 'Mycobacterial cultures') for between 2.5 and 10 generations. Final Mtb library stocks were obtained after passing the cultures through a 10-µm cell strainer (Pluriselect SKU 43-50010-03). Genomic DNA was extracted from the final stocks and library quality was validated by deep sequencing[24] (see 'Genomic DNA extraction and library preparation for Illumina sequencing of CRISPRi libraries').

## Pooled CRISPRi screen

Cultures (20 ml) were grown in vented tissue culture flasks (T-75, Falcon, 353136). 7H9 medium was supplemented with kanamycin (see 'Mycobacterial cultures') and maintained at 37 °C and 5% $CO_2$ in a humidified incubator. The screen was initiated by thawing 4× 1-ml aliquots of the Mtb Δ*bioA* (RifS or RifR mutant) CRISPRi library (RLC12) and inoculating each aliquot into 24 ml 7H9 medium supplemented with kanamycin in a T-75 flask (starting OD$_{600}$ ~0.06). The cultures were expanded to OD$_{600}$ ≈ 1.0, pooled and passed through a 10-µm cell strainer (pluriSelect, 43-50010-03) to obtain a single-cell suspension. The single-cell suspension (flow-though) was used to set up 6 'generation 0' cultures: 3 replicate cultures with ATc (+ATc) and 3 replicate control cultures without ATc (−ATc). From each generation 0 culture, we collected 10 OD$_{600}$ units of bacteria (~3 × 10$^9$ bacteria; ~30,000× coverage of the CRISPRi library) for genomic DNA extraction. The remaining culture volume was used to initiate the pooled CRISPRi fitness screen. Cultures were periodically passaged in pre-warmed medium to maintain log-phase growth. At generation 2.5, 5 and 7.5, cultures were back diluted 1:6 (to a starting OD$_{600}$ = 0.2) and cultivated for ~2.5 doublings. At generation 10, 15, 20 and 25, cultures were back diluted 1:24 (to a starting OD$_{600}$ = 0.05) and expanded for 5 generations before reaching late-log phase. ATc was replenished at every passage. By keeping the OD$_{600}$ of the 20 ml cultures ≥0.05, we guaranteed sufficient coverage of the library (3,000×) at all times. At defined timepoints (~2.5, 5, 7.5, 10, 15, 20, 25 and 30 generations), we collected bacterial pellets (10 OD$_{600}$ units) to extract genomic DNA[24]. The βH445Y library was generated and passaged in the presence of 0.05 µg ml$^{-1}$ rifampicin (2.5× the RifS MIC) to prevent reversion to a wild-type *rpoB* allele. This rifampicin concentration had no measurable impact on the growth of βH445Y. To confirm genetic stability during passaging, whole-genome sequencing was performed on one +ATc and one −ATc replicate at 3 timepoints (0, 15 and 30 generations) for the βS450L, βD435V and βH445Y libraries. These analyses confirmed that

the RifR libraries neither reverted to RifS nor acquired compensatory mutations in *rpoA*, *rpoC*, *rpoB* or *nusG*.

## Pooled CRISPRi chemical-genetic screening

Chlorflavonin chemical-genetic screening was performed as described in ref. 34. The chemical-genetic screen was initiated by thawing a 1-ml aliquot of the above Δ*bioA* (RifS or βS450L) CRISPRi library (RLC12; Addgene, 163954) and inoculating into 5 ml 7H9 supplemented (approximate starting OD of 0.2) with kanamycin (10 µg ml⁻¹) in a vented tissue culture flask (T-25, Falcon, 353109). Cultures were grown for 3 days and then expanded to 2 × 7.5 ml cultures. These cultures were grown to $OD_{600}$ ~1.0, pooled and passed through a 10-µm cell strainer (pluriSelect, 43-50010-03) to obtain a single-cell suspension. The single-cell suspension was then treated with ATc (100 ng ml⁻¹ final concentration) to initiate target pre-depletion. To generate a 5-day pre-depletion culture, the culture was diluted to a starting $OD_{600}$ of 0.1 in 40 ml with 100 ng ml⁻¹ ATc. To initiate the chemical-genetic screen, we first collected 10 $OD_{600}$ units of bacteria (~3 × 10⁹ bacteria; ~30,000× coverage of the CRISPRi library) from the 5 day CRISPRi library pre-depletion cultures as input controls. Triplicate cultures were then inoculated at $OD_{600}$ = 0.05 in 10 ml 7H9 supplemented with ATc (100 ng ml⁻¹), kanamycin (10 µg ml⁻¹) and the indicated drug concentration (Extended Data Fig. 3a) or dimethyl sulfoxide (DMSO) vehicle control. Pooled CRISPRi chemical-genetic screens were performed in vented tissue culture flasks (T-25, Falcon, 353109). Cultures were outgrown for 14 days at 37 °C and 5% $CO_2$. ATc was replenished at 100 ng ml⁻¹ at day 7. After 14 days outgrowth, $OD_{600}$ values were measured for all cultures to empirically determine the MIC for each drug. Samples from two descending doses of partially inhibitory drug concentrations were processed for genomic DNA extraction and for next generation sequencing (NGS). Concentrations were chosen to best match the degree of growth inhibition compared to the vehicle control for RifS and βS450L libraries: RifS high = 0.85 µg ml⁻¹ (75.6 ± 4.4% growth), low = 0.425 µg ml⁻¹ (97.8 ± 4.7% growth), βS450L high = 1.1 µg ml⁻¹ (57.1 ± 3.1% growth), low = 0.55 µg ml⁻¹ (107.5 ± 7.7% growth).

## Genomic DNA extraction and library preparation for Illumina sequencing of CRISPRi libraries

Genomic DNA was isolated from bacterial pellets using the CTAB–lysozyme method described previously[64], or using a modified version of the mechanical lysis method described in ref. 65. Briefly, bacterial pellets were resuspended in 600 µl TE buffer (10 mM Tris, 1 mM EDTA pH 8.0) in 2 ml tubes containing lysing matrix B (MP Biomedicals, 116911050). One volume of phenol–chloroform–isoamyl alcohol (PCI) (25:24:1) was added and samples were lysed with the Precellys Evolution homogenizer (Bertin, 02520-300-RD000) at 10,000 r.p.m. for 2 × 30 s with 5 min chilling on a freezer block or ice. After removal of the beads by spinning samples at ~12,000 × *g* for 10 min at 4 °C, the aqueous phase was transferred to a new tube containing 50 µl 5 M NaCl. After mixing by inversion, one volume of PCI was added, and samples were incubated at room temperature for ~30 min. Samples were spun at ~12,000 × *g* for 10 min at 4 °C, and the aqueous phase was transferred to a new tube. RNAse A (3 µl; Thermo Scientific, EN0531) was added and samples were incubated at 37 °C for 45 min. One volume of chloroform was added, followed by mixing with vigorous shaking. Samples were allowed to sit for 10 min. Samples were spun at  ~12,000 × *g* for 10 min at 4 °C, and the aqueous phase was transferred to a new tube with 1/10 volume 3 M sodium acetate pH 5.2 (NaOAc). One volume of cold isopropanol was added, and samples were incubated at −20 °C for 1–4 days before spinning at ~12,000 × *g* for 45 min at 4 °C. Genomic DNA was washed 2× with freshly prepared cold 70% ethanol before resuspending in 50 µl DNAse and RNAse-free water. Genomic DNA concentration was quantified using the DeNovix dsDNA high sensitivity assay (KIT-DSDNA-HIGH-2, DS-11 Series spectrophotometer/fluorometer).

To construct Illumina libraries, the sgRNA-encoding region was amplified from 500 ng genomic DNA using NEBNext Ultra II Q5 master Mix (NEB M0544L). PCR cycling conditions were: 98 °C for 45 s; 17 cycles of 98 °C for 10 s, 64 °C for 30 s, 65 °C for 20 s; 65 °C for 5 min. Each PCR reaction contained a unique indexed forward primer (0.5 µM final concentration) and a unique indexed reverse primer (0.5 µM) (Supplementary Table 5). Forward primers contained a P5 flow-cell attachment sequence, a standard Read1 Illumina sequencing primer binding site, custom stagger sequences to ensure base diversity during Illumina sequencing, and unique barcodes to allow for sample pooling during deep sequencing. Reverse primers contained a P7 flow-cell attachment sequence, a standard Read2 Illumina sequencing primer binding site and unique barcodes. Following PCR amplification, each ~230 bp amplicon was purified using AMPure XP beads (Beckman Coulter, A63882) using two-sided selection (0.75× and 0.12×). Eluted amplicons were quantified with a Qubit 2.0 Fluorometer (Invitrogen), and amplicon size and purity were quality controlled by visualization on an Agilent 4200 Tape Station (Instrument: Agilent, G2991AA; reagents: Agilent, 5067-5583; tape: Agilent, 5067-5582). Next, individual PCR amplicons were multiplexed into 20–50 nM pools and sequenced on an Illumina sequencer according to manufacturer instructions. To increase sequencing diversity, a PhiX spike-in of 2.5–10% was added to the pools (PhiX sequencing control v3; Illumina, FC-110-3001). Samples were run on the Illumina NextSeq 500, NovaSeq 6000 and NovaSeqX Plus platforms (single-read 1 × 85 cycles, 8 × i5 index cycles and 8 × i7 index cycles)[24,34].

## Differential vulnerability analysis of Rif-resistant versus Rif-sensitive strains

Gene vulnerability in the RifS and RifR *M. tuberculosis* strains was determined using an updated version of our previously described vulnerability model[24]. In the updated model, read counts for a given sgRNA in the −ATc conditions were modelled using a negative binomial distribution with a mean proportional to the counts in the +ATc condition, plus a factor representing the $\log_2$(fold change):

$$y_i^{-ATc} \sim \text{NegBinom}(\eta_i, \phi) \tag{1}$$

$$\eta_i = \log\left(y_i^{+ATc} + \lambda_i\right) + \text{TwoLine}(x_i, \alpha_l, \beta_l, \gamma, \beta_e) \tag{2}$$

where $\lambda_i$ is an sgRNA-level correction factor estimated by the model, $x_i$ represents the generations analysed for the *i*th guide, and the Two-Line function represents the piecewise linear function previously described, which models sgRNA behaviour over time. The logistic function describing gene-level vulnerabilities was simplified by setting the top asymptote of the curve (previously '*K*') equal to 0, representing the fact that weakest possible sgRNAs are expected to impose no effect on bacterial fitness, that is:

$$\text{Logistic}(s) = \frac{\beta_{max}}{\left(1 + e^{(-H \cdot (s-M))}\right)} \tag{3}$$

The Bayesian vulnerability model was run for each condition independently, and samples for all the parameters were obtained using Stan running 4 independent chains with 1,000 warmup iterations and 3,000 samples each (for a total of 12,000 posterior samples for each parameter in the model after discarding warmup iterations).

Differential vulnerabilities were estimated using two approaches. First, for each gene, the difference in pairwise (guide-level) vulnerability estimates was obtained, resulting in posterior samples of the differential vulnerability (delta vulnerability). This effectively estimated the difference in the integrals of the vulnerability functions. If the 95% credible region did not overlap 0, those were taken as significant

differential vulnerabilities between the strains. Next, to identify differences between genes that may not exhibit the expected dose–response curve, we estimated the fitness cost ($\log_2$FC) predicted by our model for a (theoretical) sgRNA of strength 0 (that is, Logistic ($s = 0$)). This represented the weakest phenotype theoretically possible with our CRISPRi system, which we call '$F_{min}$'. The difference between these $F_{min}$ values was estimated for each gene ($\Delta F_{min}$) and those where the 95% credible region did not overlap 0 were identified as significant differential vulnerabilities by this approach.

## Single-stranded (ss)DNA recombineering and validation of strains

Attenuator disruption mutations were introduced into RifS and βS450L Mtb using oligo-mediated (ssDNA) recombineering as described in ref. 63. Briefly, 70-mer oligos were designed to correspond to the lagging strand of the replication fork, with the desired mutation in the middle of the sequence. Alterations were chosen to avoid recognition by the NucS mismatch-repair machinery. RecT expression was induced ~16 h before transformation by addition of ATc to a final concentration of 0.5 μg ml$^{-1}$. Competent cells (400 μl) were transformed with 5 μg of mutation containing oligo and 0.1 μg of hygromycin resistance cassette repair oligo (1:50 ratio of mutant oligo to repair oligo) and recovered in 5 ml 7H9 media.

After 24 h of recovery, 200 μl of cells were plated on 7H10 plates supplemented with hygromycin. After 21 days of outgrowth, 12 colonies per construct were picked into 100 μl 7H9 media supplemented with hygromycin in a 96-well plate (Fisher Scientific, 877217). A volume of 50 μl of culture were heat inactivated at 80 °C for 2 h in a sealed microamp 96-well plate (Fisher Scientific, 07200684; Applied Biosystems, N8010560). A volume of 50 μl of heat-inactivated culture was mixed with 50 μl 25% DMSO and lysed at 98 °C for 10 min. Mutations of interest were confirmed by PCR amplification and Sanger sequencing. The region of interest was PCR amplified with NEBNext High-Fidelity 2× PCR Master Mix (NEB, M0541L) using 0.5 μl of heat-lysed product with the appropriate primers, annealing temperatures and extension times (see Supplementary Table 5). Amplicons were then submitted for Sanger sequencing.

## RNA extraction and RT–qPCR

Approximately 2 OD$_{600}$ units of bacteria were added to an equivalent volume of GTC buffer (5 M guanidinium thiocyanate, 0.5% sodium *N*-lauroylsarcosine, 25 mM trisodium citrate dihydrate and 0.1 M 2-mercaptoethanol), pelleted by centrifugation, resuspended in 1 ml TRIzol (Thermo Fisher, 15596026) and lysed by zirconium bead beating (MP Biomedicals, 116911050). Chloroform (0.2 ml) was added to each sample and phases were separated by centrifugation. The aqueous phase was then purified using a Direct-zol RNA miniprep kit (Zymo, R2052). Residual genomic DNA was removed by TURBO DNase treatment (Invitrogen Ambion, AM2238). After RNA cleanup and concentration (Zymo, R1017), 3 μg RNA per sample was reverse transcribed into complementary (c)DNA with random hexamers (Thermo Fisher, 18–091-050) following manufacturer instructions. RNA was removed by alkaline hydrolysis, and cDNA was purified with PCR cleanup columns (Qiagen, 28115)[24]. Next, changes in target gene expression were quantified using *Taq*Man fluorescent dye-based quantitative real-time PCR (Thermo Fisher, 4444557) on a Quantstudio System 5 (Thermo Fisher, A28140) with gene-specific qPCR primers (10 μM) and probe (10 μM), normalized to sigA (rv2703) and quantified by the $\Delta\Delta C_t$ algorithm. All gene-specific qPCR primers and probes were designed using the PrimerQuest tool from IDT (https://www.idtdna.com/PrimerQuest/Home/Index) and then validated for efficiency and linear range of amplification using standard qPCR approaches. Specificity was confirmed for each validated qPCR primer pair through melting curve analysis. All qPCR primers used in this study can be found in Supplementary Table 5.

## SEnd-seq

Total RNA extraction was performed as described above, without the use of RNA miniprep or clean and concentrator columns to preserve low abundance and short RNA transcripts. Instead, after chloroform addition, the aqueous phase was mixed 1:1 with 100% isopropanol and placed at −20 °C for 2 h, then centrifuged at -12,000 × $g$ for 15 min at 4 °C. The pellet was washed twice with 1 ml 75% (v/v) ethanol, air dried for 5 min and dissolved in nuclease-free water[36]. Subsequently, the samples were treated with 0.5 μl TURBO DNase (Invitrogen Ambion, AM2238) at 37 °C for 15 min. The RNA was then purified twice using chloroform:isoamyl alcohol (Thermo Fisher, 15593031) and recovered by ethanol precipitation.

To construct SEnd-seq libraries, 5 μg of total RNA was mixed with pooled spike-in RNAs at a mass ratio of 300:1 in a total volume of 12 μl[66]. The RNA sample was incubated with a 5′-adaptor ligation mix (1 μl 100 μM 5′ adaptor (Supplementary Table 5), 0.5 μl 50 mM ATP, 2 μl DMSO, 5 μl 50% PEG8000, 1 μl RNase Inhibitor (New England BioLabs, M0314) and 1 μl High Concentration T4 RNA Ligase 1 (New England BioLabs, M0437)) at 23 °C for 5 h. The sample was then diluted to 40 μl with nuclease-free water and cleaned twice with 1.5× volume of Agencourt RNAClean XP beads (Beckman Coulter, A63987). Immediately following the 5′ adaptor ligation, the eluted RNA was ligated to the 3′ adaptor (Supplementary Table 5) using the same procedure. After incubation at 23 °C for 5 h, the reaction was diluted to 40 μl with water and cleaned twice with 1.5× volume of Agencourt RNAClean XP beads to remove excess adaptors. The sample was subsequently eluted with 0.1× TE buffer and subjected to ribosomal RNA removal with RiboMinus Transcriptome Isolation kit (Thermo Fisher, K155004) following manufacturer instructions. After recovery by ethanol precipitation, the RNA was reverse transcribed to cDNA with *Eubacterium rectale* maturase (recombinantly purified from Eco[67], obtained from A. M. Pyle) and 5′-phosphorylated and biotinylated reverse transcription primer (Supplementary Table 5). After purification, the cDNA was circularized with TS2126 RNA ligase[68] (obtained from K. Ryan). Double-stranded DNA was synthesized using DNA Polymerase I (New England BioLabs, M0209S). After enzyme inactivation and DNA purification with 1.5× volume of AMPure beads (Beckman Coulter, A63882), the DNA was subsequently fragmented by dsDNA Fragmentase (New England BioLabs, M0348S) at 37 °C for 15 min. The reaction was stopped by adding 5 μl 0.5 M EDTA and incubated at 65 °C for 15 min in the presence of 50 mM dithiothreitol. Next, the DNA was diluted to 40 μl with TE buffer and purified with 1.5× volume of AMPure beads. The eluted DNA was used for sequencing library preparation with NEBNext Ultra II DNA Library Prep kit (New England BioLabs, E7645). Biotinylated DNA fragments were enriched with 5 μl Dynabeads M-280 Streptavidin (Thermo Fisher, 11205D) and further amplified for 12 cycles by PCR.

Following PCR amplification, each amplicon was cleaned by 1× volume of AMPure XP beads twice and quantified with a Qubit 2.0 fluorometer (Invitrogen). The amplicon size and purity were further evaluated on an Agilent 2200 Tape Station (Agilent, 5067-5576). Equal amounts of amplicon were then multiplexed and sequenced using 2 × 150 cycles on an Illumina NextSeq 500 or NovaSeq 6000 platform (Rockefeller University Genomics Resource Center)[36].

The SEnd-seq data were analysed and presented using custom analysis scripts from Ju and colleagues and is available in GitHub[36].

## LacZ and luciferase dual reporter

Mtb was cultured in standing 5 ml volumes in biological duplicate to an OD of 0.4–1.0. Fluorescein di-D-galactopyranoside (FDG) (Thermo Fisher, F1179), a cell permeable fluorescent LacZ substrate, was added to the cultures at a final concentration of 20 μM and incubated at 37 °C for 1 h. The *Renilla* Glo Assay kit from Promega (E2710) was used as instructed by the manufacturer. Briefly, after incubation, 5-ml cultures were spun down at ~3,900 × $g$ for 10 min. Pellets were resuspended in 1 ml PBS and split into two samples for parallel LacZ and *Renilla*

measurements. Samples were spun down at ~1,800 × $g$ for 10 min, LacZ samples were resuspended in 155 µl PBS, and 50 µl was plated in triplicate in a 384-well plate. Luciferase samples were resuspended in 155 µl 1× assay buffer and plated in triplicate. Fluorescence was measured at an excitation of 485 nm and an emission of 515 nm, 30 flashes and an integration time of 40 µs in a TECAN plate reader. Luminescence was measured at an integration time of 5 ms using the TECAN luminescence setting. Background signal, if applicable, was subtracted from the values. For each sample, relative expression of luciferase (RLU) values were divided by relative expression of lacZ (RFU) values. This ratio was then divided by the same value of the no-terminator construct.

## MIC$_{90}$ determinations

All compounds were dissolved in DMSO and aliquoted into a 384-well plate format using an HP D300e digital dispenser. Mtb cultures were grown to a late-logarithmic phase (OD$_{600}$ ≈ 0.8) and then back diluted to a starting OD$_{600}$ of 0.01. Of the diluted cell suspension, 50 µl was added in triplicate to wells containing the test and control compounds. Plates were incubated standing at 37 °C with 5% CO$_2$. OD$_{600}$ was measured using a Tecan Spark plate reader at days 7, 10, 12 and 14 post plating, and percent growth was calculated relative to the DMSO vehicle control for each strain. MIC$_{90}$ measurements were calculated using a nonlinear fit in GraphPad Prism. For all MIC curves, data represent the mean ± s.e.m. for triplicate wells.

## mScarlet reporter assays

mScarlet fluorescence was measured using a Tecan Spark plate reader at an excitation of 563 nm and emission of 600 nm. Mtb cultures were grown to a late-logarithmic phase (OD$_{600}$ ≈ 0.8) and then back diluted to a starting OD$_{600}$ of 0.01. A volume of 50 µl of the diluted cell suspension was added in triplicate to wells containing the test and control compounds. Plates were incubated under standing conditions and evaluated using the plate reader for fluorescence and optical density at days 5, 7, 12 and 14 post plating. Normalized fluorescence was calculated by dividing the background-adjusted fluorescence value by the background-adjusted optical density value.

## Reporting summary

Further information on research design is available in the Nature Portfolio Reporting Summary linked to this article.

## Data availability

Raw sequencing data have been deposited to the NCBI Short Read Archive under project number PRJNA1250544. Source data are provided with this paper.

## Code availability

All source code are publicly available in GitHub at https://github.com/rock-lab/collateral_vulnerabilities_rif_paper_2025 (ref. 69) and https://github.com/LiuLab-codes/SEnd_seq_analysis (ref. 70).

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

## Acknowledgements

We thank members of the Rock lab, C. Huang, M. Murray, T. Gray, K. Derbyshire, J. Wade, J. Sacchetini, E. Campbell, S. Darst and B. Landick for comments on the manuscript and/or helpful discussions; D. Schnappinger (Weill Cornell) for sharing the *ΔbioA* Mtb strain, and the Weill Cornell and Rockefeller University Genomics Core for sequencing. This work was supported by a joint NIH Tuberculosis Research Units Network (TBRU-N) grant (U19AI162584, J.M.R.), an NIH/NIAID New Innovator Award (1DP2AI144850-01, J.M.R.), a Black Family Therapeutic Development Fund (S.L.), and the Stavros Niarchos Foundation (SNF) as part of its grant to the SNF Institute for Global Infectious Disease Research at The Rockefeller University (J.M.R., S.L.).

## Author contributions

K.A.E, V.M.-G., M.A.D. and J.M.R. conceptualized the project. K.A.E., V.M.-G., S.Q.-G., M.A.D., X.J. and J.M.R. conducted investigations. K.A.E., V.M.-G., S.Q.-G., M.A.D., X.J., S.L. and J.M.R. performed data analysis. K.A.E., V.M.-G. and J.M.R. wrote the original manuscript draft. K.A.E., V.M.-G., S.Q.-G., M.A.D., X.J., S.L. and J.M.R. reviewed and edited the manuscript. S.L. and J.M.R. acquired funding and supervised the project.

## Competing interests

The authors declare no competing interests.

## Additional information

**Extended data** is available for this paper at https://doi.org/10.1038/s41564-026-02357-9.

**Correspondence and requests for materials** should be addressed to Jeremy M. Rock.

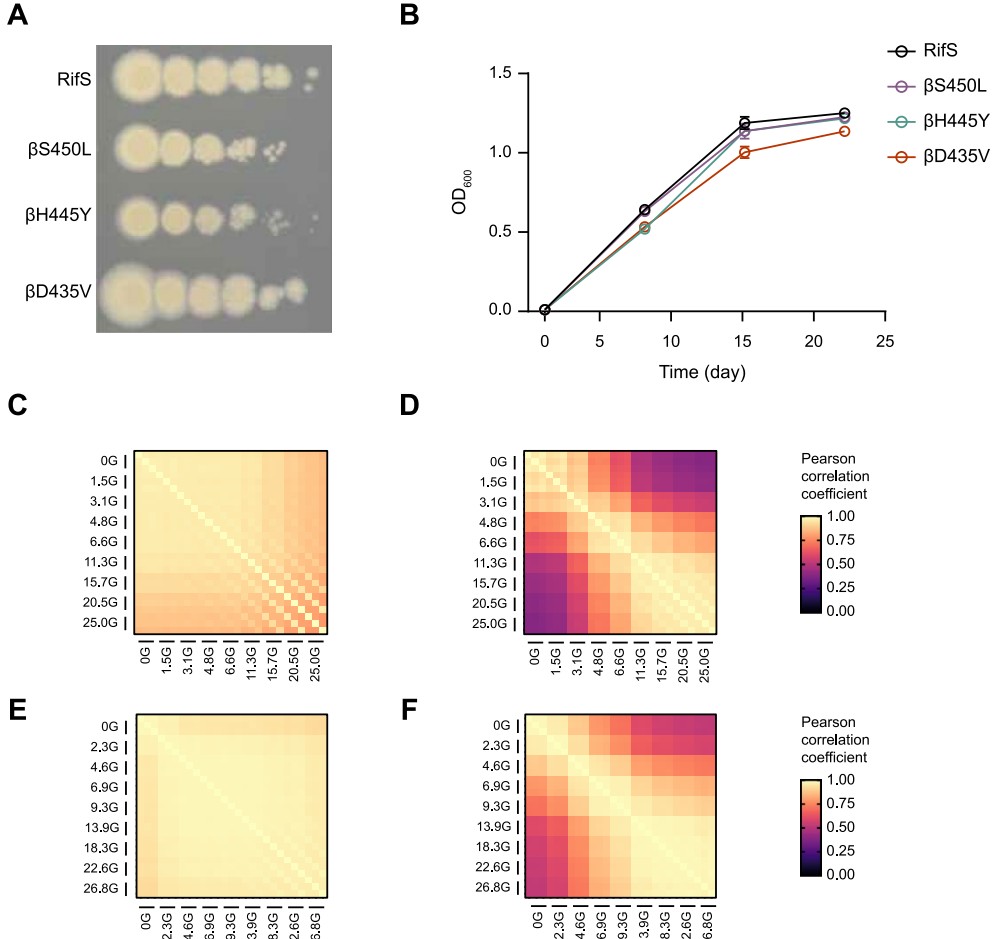

**Extended Data Fig. 1 | CRISPRi screens in RifR Mtb are well correlated.**
(**A**) Growth of 10-fold serial dilutions of ΔbioA RifS, βS450L, βD435V, βH445Y Mtb on 7H10 agar plates. Note that 7H10 media is supplemented with biotin (0.5 mg/L; ~2 μM), thereby allowing growth of the ΔbioA Mtb auxotroph. (**B**) Growth of the same strains depicted in panel (**A**) in 7H9 media. (**C-F**) Correlation heatmap of the triplicate screens depicted in Fig. 1b. Panels (**C**) and (**D**) show the

Pearson correlation between sgRNAs targeting genes predicted to be essential by TnSeq in the −ATc (**C**) or +ATc (**D**) cultures of the βD435V library. Panels (**E**) and (**F**) show the Pearson correlation between sgRNAs targeting genes predicted to be essential by TnSeq in the −ATc (**E**) or +ATc (**F**) cultures of the βH445Y library. G = generation.

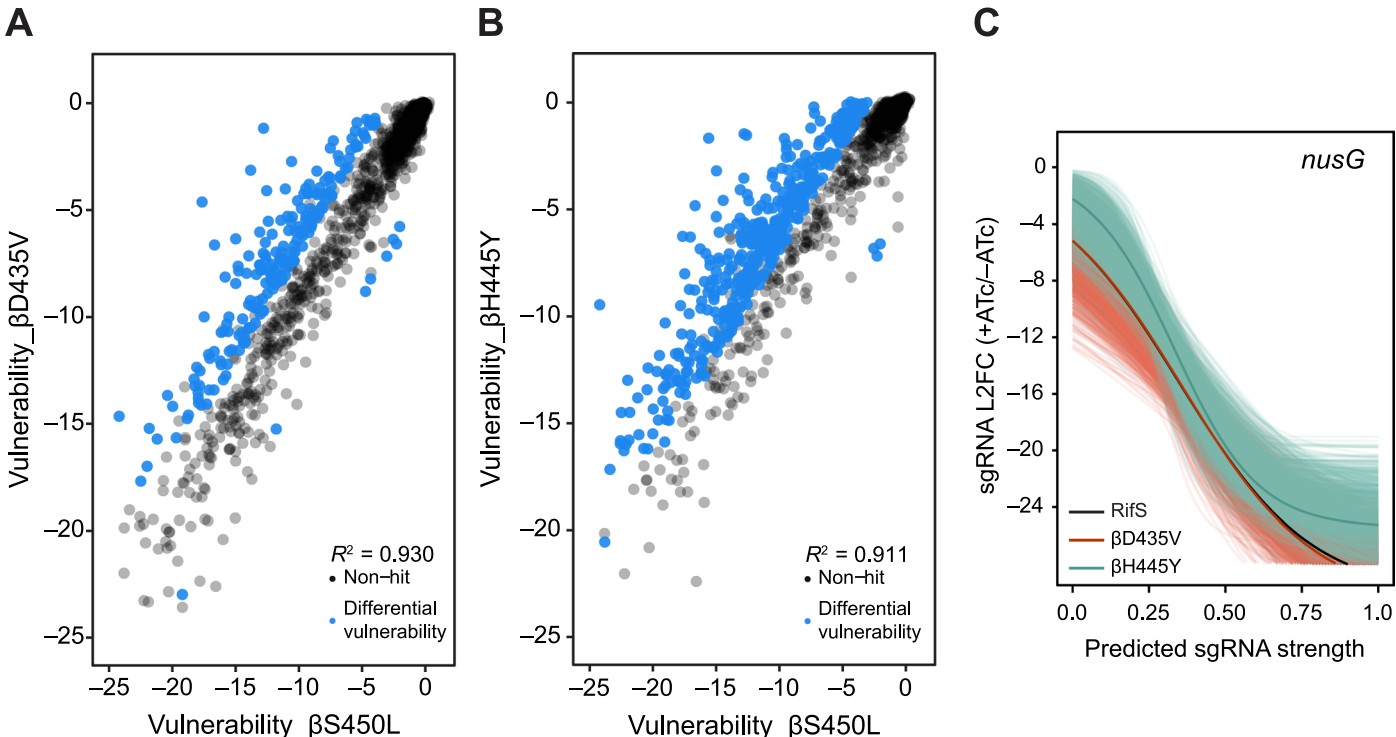

**Extended Data Fig. 2 | Differential vulnerabilities in βD435V or βH445Y Mtb.** Scatter plot showing gene vulnerability (circles) in βS450L and βD435V or βH445Y Mtb. Genes with the strongest differential vulnerabilities ($|\Delta V| \geq 3$) are highlighted in blue (**C**)*nusG* is not a collateral vulnerability in "fast" RifR RNAP mutants. Expression–fitness relationship for *nusG* in RifS (black), βD435V (orange) and βH445Y (teal). The light-colored lines represent fits generated from 1,000 samples drawn from the posterior distributions, while the dark lines indicate the mean fit.

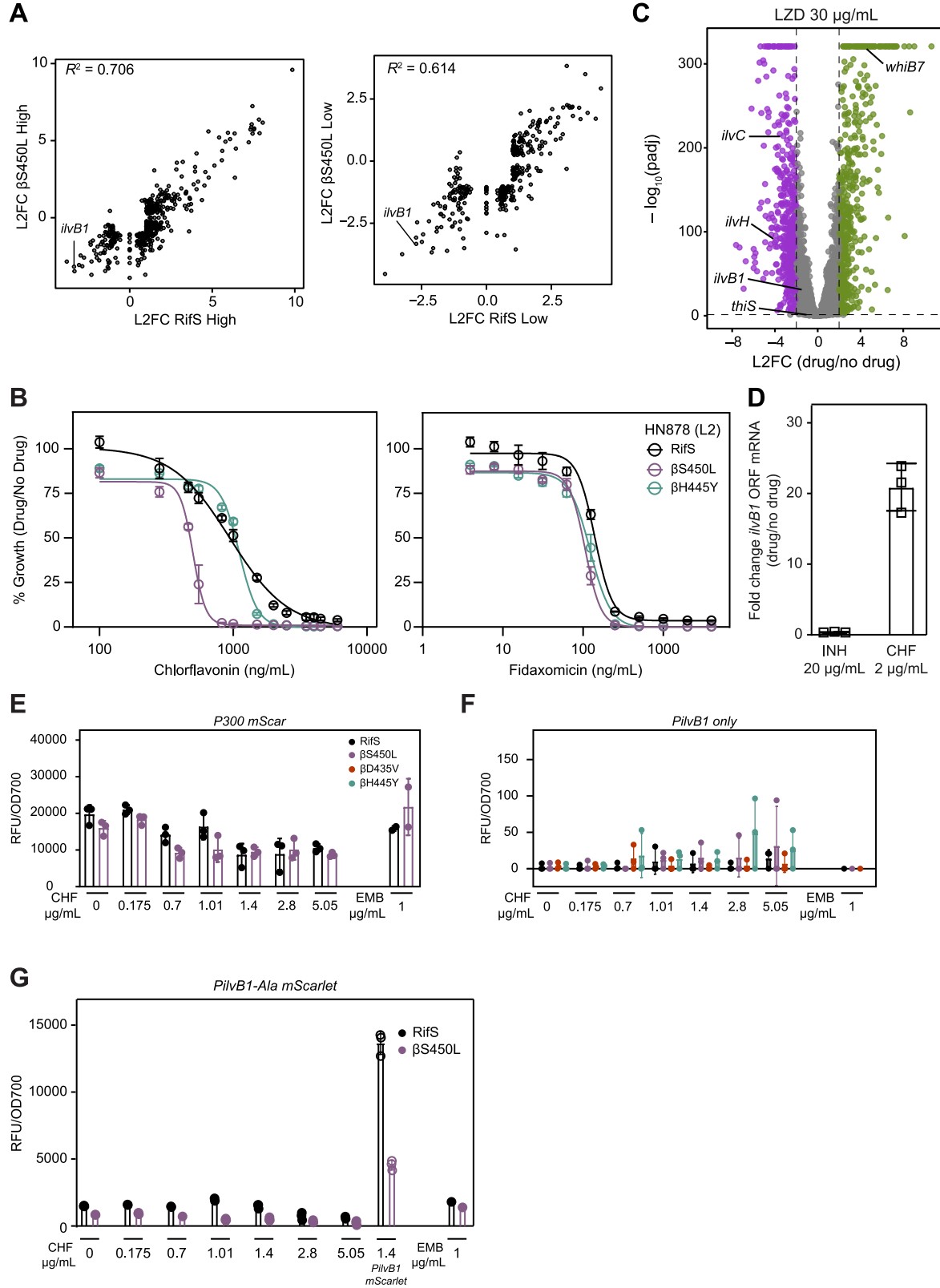

**Extended Data Fig. 3 | See next page for caption.**

**Extended Data Fig. 3 | ilvB1 is regulated by transcription attenuation in Mtb.**
(**A**)RifS and βS450L show similar chemical-genetic interaction profiles to the IlvB1 inhibitor chlorflavonin (CHF). Scatter plots depict genes (circles) quantified as significant interactors (FDR ≤ 0.05 and |L2FC | ≥ 1) in either RifS or βS450L Mtb when treated with either a high or low dose of CHF. (**B**)Dose-response curves (mean ± s.e.m., n = 3 biological replicates) for the indicated HN878 lineage 2 Mtb strains. (**C**)Differentially expressed genes when RifS Mtb is treated with the control drug linezolid (LZD) or vehicle (DMSO). Genes with an adjusted p-value less than 0.05 and an absolute log2 fold change value greater than 2, as determined by DESeq2 linear modeling[53], are shown in purple and green. (**D**)qPCR validation of the strong upregulation of *ilvB1* ORF expression when RifS Mtb is treated with CHF but not the control drug isoniazid (INH). (**E**)The mScarlet fluorescent protein expressed from the strong, constitutive P300 promoter serves as a control for the *PilvB1-mScarlet* reporter shown in Fig. 5e, f. The *P300-mScarlet* reporter was integrated into the chromosome of the four indicated Mtb strains. mScarlet fluorescence normalized to culture optical density (RFU/OD700) was monitored as a function of increasing CHF doses. Ethambutol (EMB) serves as a control drug. (**F**)*PilvB1* reporter plasmid with no mScarlet reporter. This reporter serves as the no fluorescent-protein control for for the *PilvB1-mScarlet* reporter shown in Fig. 5e, f. This control reporter was integrated into the chromosome of the four indicated Mtb strains. Autofluorescence normalized to culture optical density (RFU/OD700) was monitored as a function of increasing CHF doses. Ethambutol (EMB) serves as a control drug. (**G**)The four putative BCAA regulatory codons "LVVI" (see Fig. 5) in the mScarlet reporter described in Fig. 5e (*PilvB1-mScarlet*) were mutated to "AAAA" (*PilvB1-Ala-mScarlet*). The introduced mutations are not predicted to affect the folding of the anti-terminator or terminator hairpins. This reporter was then integrated into the chromosome of either RifS or βS450L Mtb. Reporter activity was quantified as in panel 5 F. This reporter shows a low, CHF dose-independent expression of mScarlet in both RifS and βS450L Mtb (q = 0.101 by unpaired t test).

**A**

| | M. fortuitum | M. kansasii | M. marinum | M. abscessus |
|---|---|---|---|---|
| Species | | | | |
| Leader peptide | M**LVVI**GWR**V**DAA**L**A**L**CRA* | M**LVVL**GRR**V**GA* | MDTAGTPGK**LVVL**GRR**VV**A* | **V**NNQA**LLVVI**GMR**V**DA* |
| Terminator stability | $\Delta G_{min}$=−19.72 kcal/mol | $\Delta G_{min}$=−19.70 kcal/mol | $\Delta G_{min}$=−19.80 kcal/mol | $\Delta G_{min}$=−18.90 kcal/mol |

| | M. ulcerans | M. avium | E. coli |
|---|---|---|---|
| Species | | | |
| Leader peptide | MDTARTPGK**LVVL**GRR**VV**A* | M**LVVI**RR**V**GA* | MTTSM**L**NAK**LL**PTAPSAA**VVVV**R**VVVVV**GNAP* |
| Terminator stability | $\Delta G_{min}$=−19.80 kcal/mol | $\Delta G_{min}$=−19.00 kcal/mol | $\Delta G_{min}$=−21.10 kcal/mol |

**Extended Data Fig. 4 | Predicted leader peptide sequences of ilvB1 homologs in other bacteria harbor putative BCAA-regulatory codons.** Sequence of the predicted *ilvB1* leader peptide and downstream Rho-independent terminator[51] from the indicated bacterial species. Diagrams depict the predicted RNA folding and free energy[51] of the indicated hairpin.

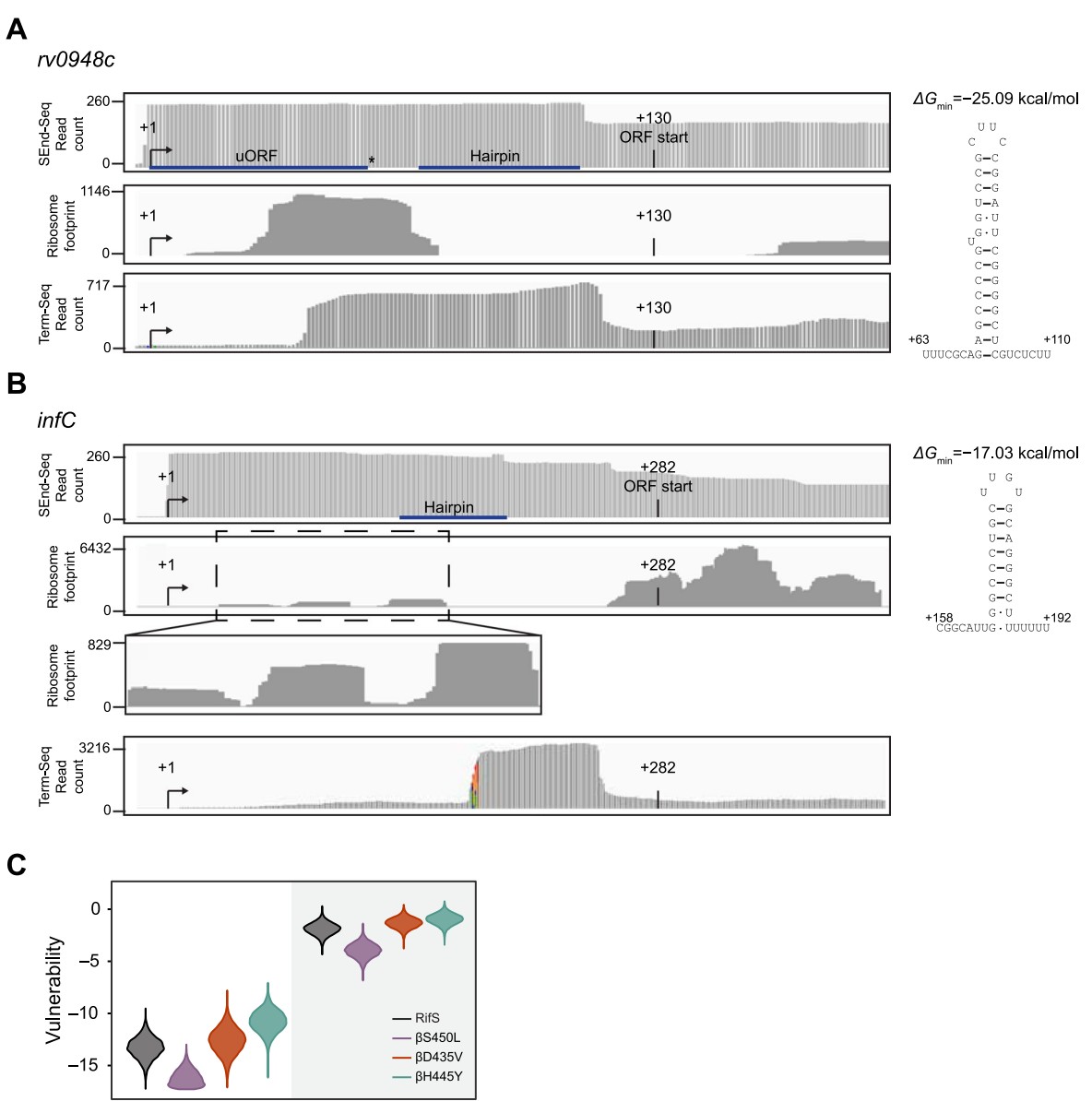

**Extended Data Fig. 5 | Transcription attenuation may contribute to multiple collateral vulnerabilities in βS450L Mtb.** Top panel shows SEnd-Seq data tracks depicting total RNA coverage of the indicated 5' leader region. +1 marks the transcription start site (TSS) and the ORF start is indicated. Middle panel shows ribosome profiling[50] data track depicting ribosome occupancy within the indicated 5' leader region. Bottom panel shows Term-Seq[38] data track depicting 3' end coverage of transcripts from log-phase Mtb. Right panels depict the predicted RNA folding and free energy[51] of the indicated hairpin. Putative uORF and regulatory hairpin regions are indicated by a blue line. (**C**) Vulnerability distributions for the genes depicted in panels **A** and **B** in the indicated Mtb strains.

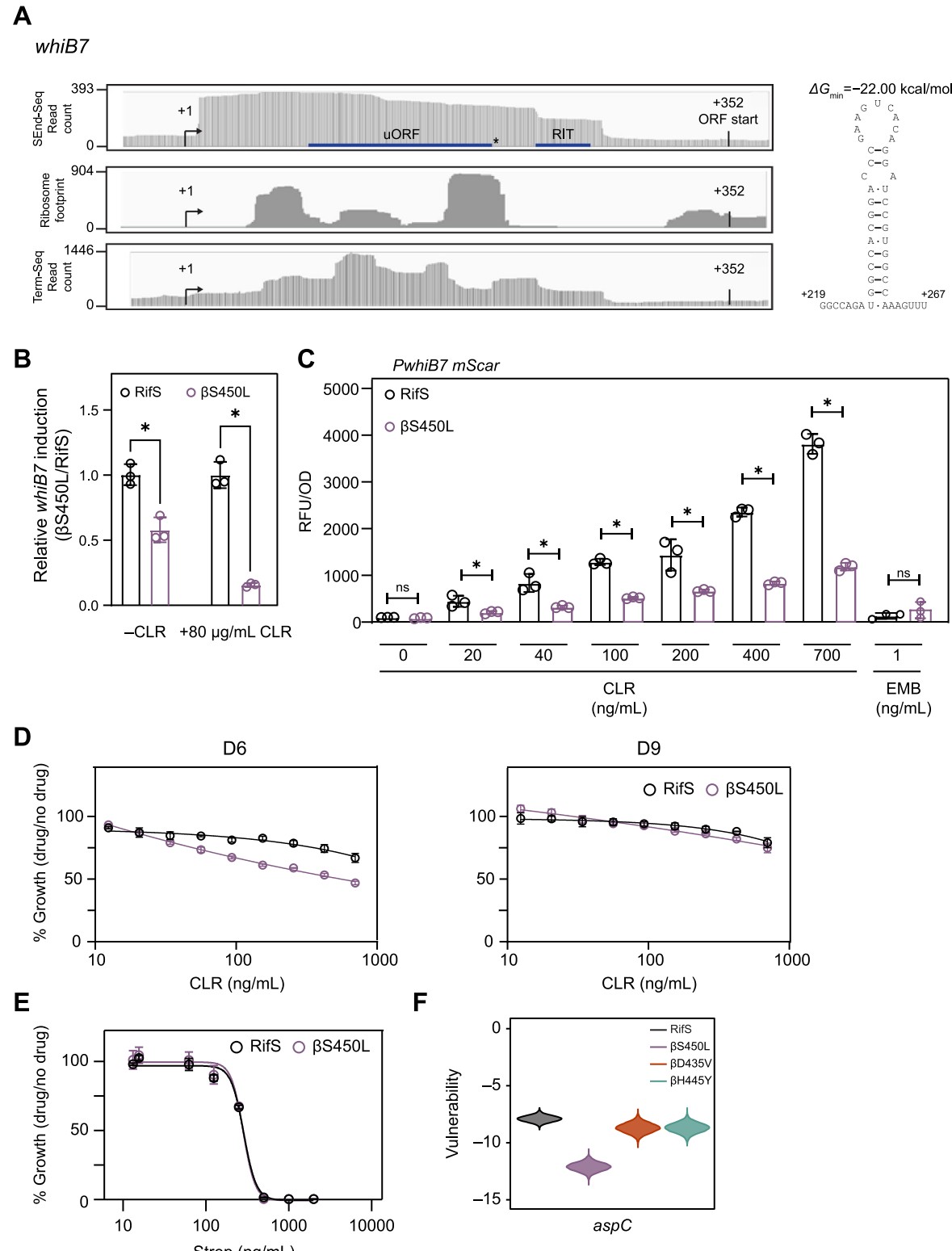

**Extended Data Fig. 6 | See next page for caption.**

**Extended Data Fig. 6 | Transcription attenuation contributes to weaker whiB7 induction in βS450L Mtb.** (**A**) Top panel shows SEnd-Seq data tracks depicting total RNA coverage of the indicated 5' leader region. +1 marks the transcription start site (TSS) and the ORF start is indicated. Middle panel shows ribosome profiling[50] data track depicting ribosome occupancy within the indicated 5' leader region. Bottom panel shows Term-Seq[38] data track depicting 3' end coverage of transcripts from log-phase Mtb. Right panel depicts the predicted RNA folding and free energy[51] of the indicated hairpin. Putative uORF and regulatory hairpin regions are indicated by a blue line. (**B**) *whiB7* ORF induction relative to RifS Mtb after 24 h of CLR treatment (right) or vehicle control (left). "*" indicates a significant difference determined by multiple unpaired multiple unpaired, two-sided Welch t-tests with multiple hypothesis correction (Benjamini, Krieger, and Yekutieli, 1% FDR). q value minus CLR is 0.0046, q value plus CLR is 0.0041. These data are derived from three technical replicates and are representative of two independent experiments. (**C**) Regulation of the *whiB7* 5' leader was monitored by fusing the *mScarlet* fluorescent protein to the *whiB7* promoter and leader sequence (*PwhiB7-mScarlet*). This reporter plasmid was then integrated into the chromosome of the two indicated Mtb strains. mScarlet fluorescence normalized to culture optical density (RFU/OD700) was monitored as a function of increasing CLR doses. Ethambutol (EMB) serves as a control drug (1 µg ml$^{-1}$ = 1X MIC$_{90}$ for RifS Mtb). "*" indicates a significant difference determined by multiple unpaired Welch t-tests with multiple hypothesis correction (Benjamini, Krieger, and Yekutieli, 5% FDR). "ns" indicates a nonsignificant difference by the same metric. These data are derived from three technical replicates and are representative of two independent experiments. (**D**) Clarithromycin (CLR) dose-response curves (mean ± s.e.m., n = 3 biological replicates) for RifS (black) and βS450L (purple) Mtb measured after 6 and 9 days of drug exposure. (**E**) Streptomycin dose-response curves (mean ± s.e.m., n = 3 biological replicates) for RifS (black) and βS450L (purple) Mtb. (**F**) Vulnerability distributions for *aspC* in the indicated Mtb strains.

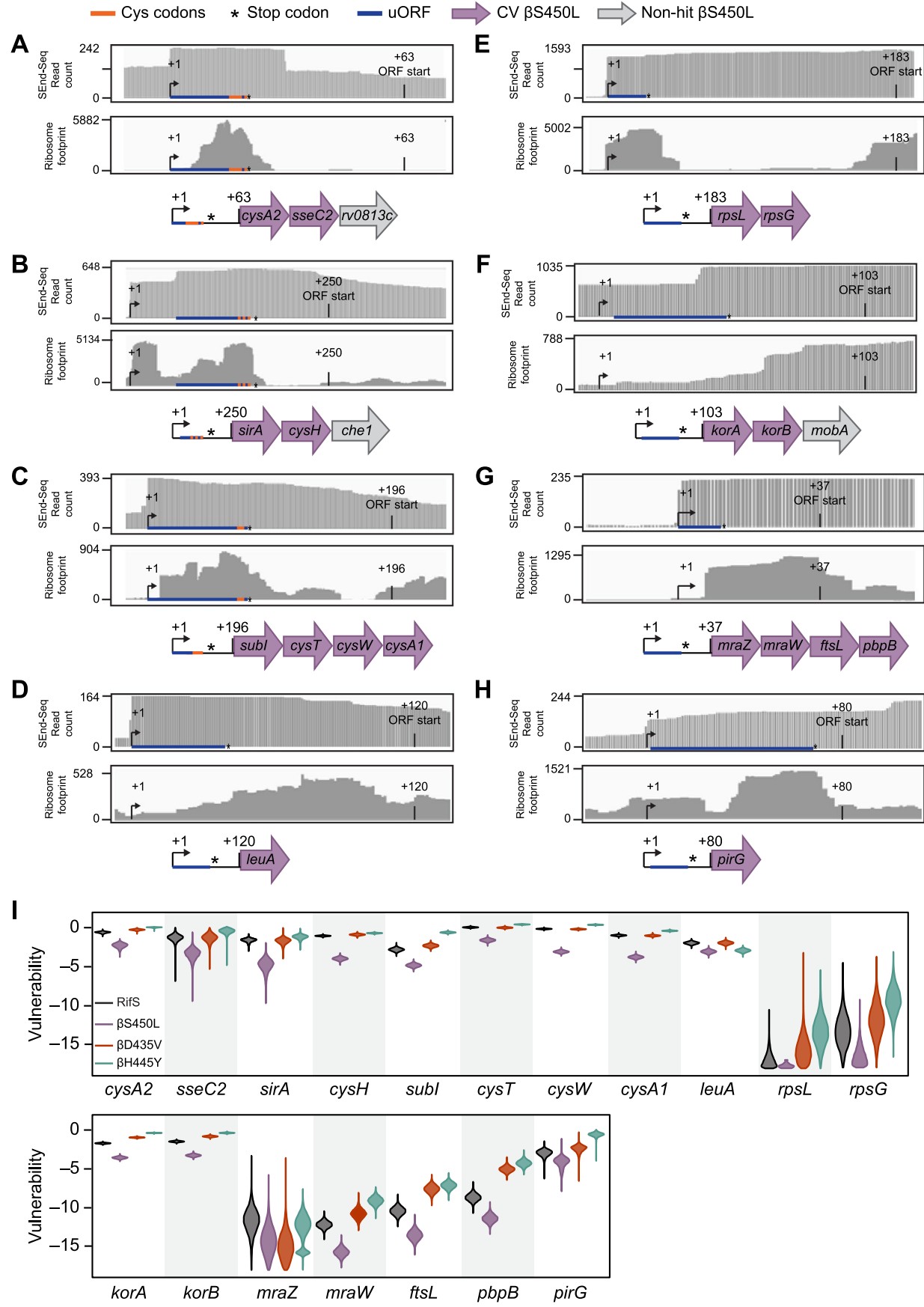

**Extended Data Fig. 7 | See next page for caption.**

**Extended Data Fig. 7 | Defective translation attenuation may drive additional collateral vulnerabilities in βS450L Mtb.** Top panel shows SEnd-Seq data track depicting total RNA coverage of the indicated 5′ leader region. +1 marks the transcription start site (TSS) and the ORF start is indicated. Bottom panel shows ribosome profiling[50] data track depicting ribosome occupancy within the indicated 5′ leader region. Putative uORF regions and regulatory cysteine codons are indicated by blue and orange lines, respectively[57]. (**I**) Vulnerability distributions for the indicated genes in the indicated Mtb strains.

# Reporting Summary

## Statistics

For all statistical analyses, confirm that the following items are present in the figure legend, table legend, main text, or Methods section.

| n/a | Confirmed | |
|---|---|---|
| ☐ | ☒ | The exact sample size ($n$) for each experimental group/condition, given as a discrete number and unit of measurement |
| ☐ | ☒ | A statement on whether measurements were taken from distinct samples or whether the same sample was measured repeatedly |
| ☐ | ☒ | The statistical test(s) used AND whether they are one- or two-sided *Only common tests should be described solely by name; describe more complex techniques in the Methods section.* |
| ☒ | ☐ | A description of all covariates tested |
| ☐ | ☒ | A description of any assumptions or corrections, such as tests of normality and adjustment for multiple comparisons |
| ☐ | ☒ | A full description of the statistical parameters including central tendency (e.g. means) or other basic estimates (e.g. regression coefficient) AND variation (e.g. standard deviation) or associated estimates of uncertainty (e.g. confidence intervals) |
| ☐ | ☒ | For null hypothesis testing, the test statistic (e.g. $F$, $t$, $r$) with confidence intervals, effect sizes, degrees of freedom and $P$ value noted *Give P values as exact values whenever suitable.* |
| ☐ | ☒ | For Bayesian analysis, information on the choice of priors and Markov chain Monte Carlo settings |
| ☐ | ☒ | For hierarchical and complex designs, identification of the appropriate level for tests and full reporting of outcomes |
| ☐ | ☒ | Estimates of effect sizes (e.g. Cohen's $d$, Pearson's $r$), indicating how they were calculated |

*Our web collection on statistics for biologists contains articles on many of the points above.*

## Software and code

Policy information about availability of computer code

Data collection    Data collection was done with custom scripts dependent on third-party tools. All source code will be made publicly available online(GitHub: https://github.com/rock-lab/ collateral_vulnerabilities_rif_paper_2025).

Data analysis    GraphPad Prism (version 10.0.2)
Microsoft Excel(365)
bwa (version 0.7.17-r1188 or v1.3.1, depending on operating system requirements)
HaplotypeCaller tool Genome Analysis Toolkit (version 3.5)
samtools (version 1.7)
Mykrobe (version 0.9.012)
Snippy9 (version 3.2-dev or v4.6.0, depending on operating system requirements)
freebayes (version 1.3.1)
Image J software (NIH)
viridis (version 0.6.3)
reshape (version 0.8.9)
Epi (version 2.47.1)
readxl (version 1.4.2)
RColorBrewer (version 1.1-3)
tidyverse (version 2.0.0)
ggplot2 (version 3.4.2)
subread-align (version 1.6.0)
Python (version 3.11.2)

SciPy (version 1.10.1)
numpy (version 1.23.5)
R (version 4.2.1)
Stan (version 2.21.8)
Rstan (version 2.21.8)

For manuscripts utilizing custom algorithms or software that are central to the research but not yet described in published literature, software must be made available to editors and reviewers. We strongly encourage code deposition in a community repository (e.g. GitHub). See the Nature Portfolio guidelines for submitting code & software for further information.

## Data

Policy information about availability of data

All manuscripts must include a data availability statement. This statement should provide the following information, where applicable:
- Accession codes, unique identifiers, or web links for publicly available datasets
- A description of any restrictions on data availability
- For clinical datasets or third party data, please ensure that the statement adheres to our policy

Raw sequencing data are deposited to the NCBI Short Read Archive under project number PRJNA1250544.

## Research involving human participants, their data, or biological material

Policy information about studies with human participants or human data. See also policy information about sex, gender (identity/presentation), and sexual orientation and race, ethnicity and racism.

| | |
|---|---|
| Reporting on sex and gender | NA |
| Reporting on race, ethnicity, or other socially relevant groupings | NA |
| Population characteristics | NA |
| Recruitment | NA |
| Ethics oversight | NA |

Note that full information on the approval of the study protocol must also be provided in the manuscript.

# Field-specific reporting

Please select the one below that is the best fit for your research. If you are not sure, read the appropriate sections before making your selection.

☒ Life sciences          ☐ Behavioural & social sciences          ☐ Ecological, evolutionary & environmental sciences

For a reference copy of the document with all sections, see nature.com/documents/nr-reporting-summary-flat.pdf

# Life sciences study design

All studies must disclose on these points even when the disclosure is negative.

| | |
|---|---|
| Sample size | The passaging and competitive growth data presented in the manuscript were conducted in biological triplicate. Previous work on libraries these size (Bosch et al., 2021 and Li et al., 2022) showed that 3 biological replicates are enough to detect statistically significant differences between conditions. |
| Data exclusions | None |
| Replication | We have indicated the number of times experiments were independently preformed in the figure legends and their corresponding methods. |
| Randomization | Strains were sequenced to determine that no other SNPs were different between the WT and RifR strains. As this was the only relevant variable, one was assigned as control (WT) and one as experimental (RifR) and no randomization was needed. |
| Blinding | Blinding was not performed for any experiments as measurements of optical density, spot assays, transcription readthrough, and mScarlet fluorescence quantification do not require researcher-based judgments and therefore blinding was not deemed necessary. |

# Reporting for specific materials, systems and methods

We require information from authors about some types of materials, experimental systems and methods used in many studies. Here, indicate whether each material, system or method listed is relevant to your study. If you are not sure if a list item applies to your research, read the appropriate section before selecting a response.

## Materials & experimental systems

| n/a | Involved in the study |
|-----|----------------------|
| ☒ ☐ | Antibodies |
| ☒ ☐ | Eukaryotic cell lines |
| ☒ ☐ | Palaeontology and archaeology |
| ☒ ☐ | Animals and other organisms |
| ☒ ☐ | Clinical data |
| ☒ ☐ | Dual use research of concern |
| ☒ ☐ | Plants |

## Methods

| n/a | Involved in the study |
|-----|----------------------|
| ☒ ☐ | ChIP-seq |
| ☒ ☐ | Flow cytometry |
| ☒ ☐ | MRI-based neuroimaging |

## Plants

| | |
|---|---|
| Seed stocks | *Report on the source of all seed stocks or other plant material used. If applicable, state the seed stock centre and catalogue number. If plant specimens were collected from the field, describe the collection location, date and sampling procedures.* |
| Novel plant genotypes | *Describe the methods by which all novel plant genotypes were produced. This includes those generated by transgenic approaches, gene editing, chemical/radiation-based mutagenesis and hybridization. For transgenic lines, describe the transformation method, the number of independent lines analyzed and the generation upon which experiments were performed. For gene-edited lines, describe the editor used, the endogenous sequence targeted for editing, the targeting guide RNA sequence (if applicable) and how the editor was applied.* |
| Authentication | *Describe any authentication procedures for each seed stock used or novel genotype generated. Describe any experiments used to assess the effect of a mutation and, where applicable, how potential secondary effects (e.g. second site T-DNA insertions, mosiacism, off-target gene editing) were examined.* |

