## [Peer Review File · Nature Microbiology]

Transcription attenuation amplifies collateral vulnerabilities in rifampicin-resistant *Mycobacterium tuberculosis*

Corresponding Author: Dr Jeremy Rock

Version 0:

Decision Letter:

12th May 2025

Dear Professor Rock,

Thank you for submitting your Article entitled "Transcription attenuation amplifies collateral vulnerabilities in rifampicin-resistant *Mycobacterium tuberculosis*" for consideration in Nature Microbiology and please accept our apologies for the time it has taken us to contact you with a decision on your manuscript, which is due to our current high submission volume. I regret to inform you that after careful discussion within the editorial team, we have decided that we cannot consider it for publication here.

As you may know, we decline a substantial proportion of manuscripts without sending them to referees, so that they may be sent elsewhere without delay. In such cases, even if referees were to certify the manuscript as technically correct, we consider that the work does not represent the type of advance that Nature Microbiology seeks to publish. This editorial assessment is based on considerations such as the degree of conceptual advance provided, the breadth of potential interest to researchers and timeliness.

In this case, although we have no doubt of the interest that your study will have to others working in this field, given the lack of demonstration that the collateral vulnerabilities can be targeted *in vivo*, in terms of the overall degree of advance and immediate relevance to our broad microbiological readership, we do not feel that the current work has quite met the high bar that we must unfortunately set for further consideration towards publication in Nature Microbiology. We therefore feel that the paper would find a more suitable outlet in another journal.

Although we cannot offer to publish your manuscript, my colleagues at Nature Communications will send your manuscript out for external review. To transfer your manuscript please use our manuscript transfer portal. You will not have to re-supply manuscript metadata and files, unless you wish to make modifications. For more information, please see our [manuscript transfer FAQ](http://www.nature.com/authors/author_resources/transfer_manuscripts.html?WT.mc_id=EMI_NPG_1511_AUTHORTRANSF&WT.ec_id=AUTHOR) page.

Please be assured that this editorial decision does not represent a criticism of the quality of your work, nor are we questioning its value to others working in this area. We hope that you will rapidly receive a more favourable response elsewhere.

I am sorry that we cannot respond more positively on this occasion.

Version 1:

Reviewer comments:

Reviewer #1

(Remarks to the Author)

Review of Eckart et al., NMICROBIOL-25051584A-Z, "Transcription attenuation amplifies collateral vulnerabilities in rifampicin-resistant *Mycobacterium tuberculosis*"

This manuscript describes studies aimed at understanding collateral vulnerabilities and resistances arising in *M. tuberculosis* as a result of acquisition of rifampicin resistance. The authors have previously described these for the most common clinically acquired rifampicin allele (BS450L) and in this manuscript describe these for the next two most common alleles. At a high level it is a bit disappointing that there was little overlap between collateral changes amongst the three different changes in RNAP. The authors focused on understanding vulnerabilities unique to BS450L and elegantly show that these vulnerabilities are a result of

enhanced termination of RNAP in a regulatory region controlling branch chain amino acid biosynthesis.

This is really solid and beautiful work, and I had only a few minor comments:

(1) I wanted to dig in and see the full list of the highest differentially vulnerable amongst the three mutants (the blue dots in Fig 2) but the only place this is shown is in the source data in Supplemental Table 1. A full multi-tab 4000 gene complex dataset is pretty serious overkill for this, maybe a short table could be added (or even a tab that ties back to Figure 2) as I am certain others will want easy access to that gene list.

(2) In the discussion the authors may want to point out that these allele-specific collateral changes may not be applicable to all kinds of targets, hopefully some metabolic targets will have collateral sensitivities that are more conducive to engineering combinations that could suppress emergence of resistance.

Reviewer #2

(Remarks to the Author)

Drug-resistance, and in particular Rifampin resistance in Mtb is a major global health threat. However, work in cancer as well as infectious diseases has identified that drug-resistant cells can be more susceptible to other agents – i.e. collateral vulnerability. In this work, Eckart and colleagues build on the recent work from the Rock group that identified collateral vulnerabilities of the most common Rif^R mutation, RpoB-S450L. Here, they perform a similar experiment as performed previously for that strain, but with two other Rif^R mutant strains – in which the mutations cause faster as opposed to slower RNAP in an H37Rv background. Intriguingly, there are no (or v. few – it's unclear) shared collateral vulnerabilities between all 3 strains. It's not even clear if the two 'fast RNAP' strains share any vulnerabilities. Following up on S3450L vulnerabilities, the authors focus on branched-chain aa synthesis genes, esp ilBb1. Genetic or chemical inhibition of this gene leads to greater growth attenuation in the S450L strain compared with its WT parent. Further focusing the mechanism of this vulnerability, they confirm that it seems that ilvB1 and its operon is regulated similarly to other model bacteria – i.e. there is a regulatory region upstream of the ORF that codes for a sORF that has a short stretch enriched for BCAA.

Overall, this is an elegant and well-performed study that confirms and follows up on a hit from the previous work of the group and shows that this vulnerability is 'unique' to this specific Rif^R mutation.

Comments

1. Fig. 1: it seems that the H445Y strain has a very few vulnerabilities. I know that the data are in the supplement, but it's not readily apparent if any of the hits are shared with D435V. Can the authors call out the few hits (in blue) and explicitly clarify this point?
2. The lack of potential or major overlap between the two 'fast RNAP' strains is surprising. While it didn't become the major focus of this work, can the authors speculate more about this in the Discussion?
3. And just to confirm, where there any shared hits between all three Rif^R strains?
4. Fig. 4E: this experiment should be performed with the two other Rif^R strains, not just S450L, to confirm that S450L is the only one that is more sensitive to this drug.
5. The proposed mechanism involves translation of a BCAA-rich region in cryptic uORF in the 5' UTR of the operon. If this is the case, one might predict that S450L Rif^R strains are more susceptible to leuRS inhibition also (which are commercially available). Although the mutation of the Rif^R is not stated, in this publication (PMID: 27503647), that did not seem to be the case. Could the authors test this agent or another LeuRS inhibitor with the 3 Rif^R strains and WT parent?
6. Although the genetic architecture of this operon is conserved, strain background may influence metabolic vulnerabilities. The authors should discuss this, or ideally, test the genetic and chemical collateral vulnerabilities in at least one non-lineage 4 strain (Rif^R/ Rif^S) strain.

Reviewer #3

(Remarks to the Author)

Decision Letter:

25th June 2025

Dear Jeremy,

Thank you for your patience while your manuscript "Transcription attenuation amplifies collateral vulnerabilities in rifampicin-resistant Mycobacterium tuberculosis" was under peer-review at Nature Microbiology. It has now been seen by 3 referees,

whose expertise and comments you will find at the end of this email. Although they find your work of some potential interest, they have raised a number of concerns that will need to be addressed before we can consider publication of the work in Nature Microbiology.

In particular, referee #1 suggests adding a short table with a gene list, and to discuss some of the limitations of the study. Referee #2 asks for some more discussion of lack of potential or major overlap between the two 'fast RNAP' strains. Furthermore, this referee says the experiment in Fig. 4E should be performed with the two other Rif^R strains to confirm that S450L is the only one that is more sensitive to this drug. Referee #2 also suggests to test LeuRS inhibition with the 3 Rif^R strains and WT parent. Ideally, you should test the genetic and chemical collateral vulnerabilities in at least one non-lineage 4 strain (Rif^R/ Rif^S) strain. Referee #3 (please see report attached) suggests more comprehensive classification of genes into three categories, asks why so many genes appear to provide a growth advantage in β H445Y, and says further clarification or validation is needed. The referee is also concerned that the effectiveness of targeted interventions could be limited in β S450L M. tuberculosis strains carrying such mutations. Further, referee #3 says that the observed phenotype might differ in vivo, and that at the very least, this should be discussed. Furthermore, referee #3 says it would be helpful to express ilvB1 under the control of a constitutive promoter to determine whether transcription attenuation alone accounts for the collateral vulnerability of ilvB1 in the β S450L strain. Editorially, we will require these concerns to be addressed in full.

Should further experimental data allow you to address these criticisms, we would be happy to look at a revised manuscript.

Please include a data availability statement as a separate section after Methods but before references, under the heading "Data Availability". This section should inform readers about the availability of the data used to support the conclusions of your study. This information includes accession codes to public repositories (data banks for protein, DNA or RNA sequences, microarray, proteomics data etc...), references to source data published alongside the paper, unique identifiers such as URLs to data repository entries, or data set DOIs, and any other statement about data availability. At a minimum, you should include the following statement: "The data that support the findings of this study are available from the corresponding author upon request", mentioning any restrictions on availability. If DOIs are provided, we also strongly encourage including these in the Reference list (authors, title, publisher (repository name), identifier, year). For more guidance on how to write this section please see: <http://www.nature.com/authors/policies/data/data-availability-statements-data-citations.pdf>

* If you have not done so already we suggest that you begin to revise your manuscript so that it conforms to our Article format instructions at <http://www.nature.com/nmicrobiol/info/final-submission>. Refer also to any guidelines provided in this letter.

When submitting the revised version of your manuscript, please pay close attention to our [href="https://www.nature.com/nature-portfolio/editorial-policies/image-integrity">Digital Image Integrity Guidelines. and to the following points below:](https://www.nature.com/nature-portfolio/editorial-policies/image-integrity)

EXTENDED DATA FIGURES

Link Redacted

Note: This url links to your confidential homepage and associated information about manuscripts you may have submitted or be reviewing for us. If you wish to forward this e-mail to co-authors, please delete this link to your homepage first.

Nature Microbiology is committed to improving transparency in authorship. As part of our efforts in this direction, we are now requesting that all authors identified as 'corresponding author' on published papers create and link their Open Researcher and Contributor Identifier (ORCID) with their account on the Manuscript Tracking System (MTS), prior to acceptance. This applies to primary research papers only. ORCID helps the scientific community achieve unambiguous attribution of all scholarly contributions. You can create and link your ORCID from the home page of the MTS by clicking on 'Modify my Springer Nature account'. For more information please visit www.springernature.com/orcid.

If you wish to submit a suitably revised manuscript we would hope to receive it within 4 months.

Reviewer Expertise:

Referee #1: Mtb, TB

Referee #2: Mtb rifampicin-resistance

Referee #3: CRISPR-based screens, Mtb

Reviewer Comments:

Reviewer #1 (Remarks to the Author):

Review of Eckart et al., NMICROBIOL-25051584A-Z, "Transcription attenuation amplifies collateral vulnerabilities in rifampicin-resistant *Mycobacterium tuberculosis*"

This manuscript describes studies aimed at understanding collateral vulnerabilities and resistances arising in *M. tuberculosis* as a result of acquisition of rifampicin resistance. The authors have previously described these for the most common clinically acquired rifampicin allele (BS450L) and in this manuscript describe these for the next two most common alleles. At a high level it is a bit disappointing that there was little overlap between collateral changes amongst the three different changes in RNAP. The authors focused on understanding vulnerabilities unique to BS450L and elegantly show that these vulnerabilities are a result of enhanced termination of RNAP in a regulatory region controlling branch chain amino acid biosynthesis.

This is really solid and beautiful work, and I had only a few minor comments:

(1) I wanted to dig in and see the full list of the highest differentially vulnerable amongst the three mutants (the blue dots in Fig 2) but the only place this is shown is in the source data in Supplemental Table 1. A full multi-tab 4000 gene complex dataset is pretty serious overkill for this, maybe a short table could be added (or even a tab that ties back to Figure 2) as I am certain others will want easy access to that gene list.

(2) In the discussion the authors may want to point out that these allele-specific collateral changes may not be applicable to all kinds of targets, hopefully some metabolic targets will have collateral sensitivities that are more conducive to engineering combinations that could suppress emergence of resistance.

Reviewer #2 (Remarks to the Author):

Drug-resistance, and in particular Rifampin resistance in *Mtb* is a major global health threat. However, work in cancer as well as infectious diseases has identified that drug-resistant cells can be more susceptible to other agents – i.e. collateral vulnerability. In this work, Eckart and colleagues build on the recent work from the Rock group that identified collateral vulnerabilities of the most common *RifR* mutation, *RpoB*-S450L. Here, they perform a similar experiment as performed previously for that strain, but with two other *RifR* mutant strains – in which the mutations cause faster as opposed to slower RNAP in an H37Rv background. Intriguingly, there are no (or v. few – it's unclear) shared collateral vulnerabilities between all 3 strains. It's not even clear if the two 'fast RNAP' strains share any vulnerabilities. Following up on S3450L vulnerabilities, the authors focus on branched-chain aa synthesis genes, esp *ilvB1*. Genetic or chemical inhibition of this gene leads to greater growth attenuation in the S450L strain compared with its WT parent. Further focusing the mechanism of this vulnerability, they confirm that it seems that *ilvB1* and its operon is regulated similarly to other model bacteria – i.e. there is a regulatory region upstream of the ORF that codes for a sORF that has a short stretch enriched for BCAA.

Overall, this is an elegant and well-performed study that confirms and follows up on a hit from the previous work of the group and

shows that this vulnerability is 'unique' to this specific RifR mutation.

Comments

1. Fig. 1: it seems that the H445Y strain has a very few vulnerabilities. I know that the data are in the supplement, but it's not readily apparent if any of the hits are shared with D435V. Can the authors call out the few hits (in blue) and explicitly clarify this point?
2. The lack of potential or major overlap between the two 'fast RNAP' strains is surprising. While it didn't become the major focus of this work, can the authors speculate more about this in the Discussion?
3. And just to confirm, where there any shared hits between all three RifR strains?
4. Fig. 4E: this experiment should be performed with the two other RifR strains, not just S450L, to confirm that S450L is the only one that is more sensitive to this drug.
5. The proposed mechanism involves translation of a BCAA-rich region in cryptic uORF in the 5' UTR of the operon. If this is the case, one might predict that S450L RifR strains are more susceptible to leuRS inhibition also (which are commercially available). Although the mutation of the RifR is not stated, in this publication (PMID: 27503647), that did not seem to be the case. Could the authors test this agent or another LeuRS inhibitor with the 3 RifR strains and WT parent?
6. Although the genetic architecture of this operon is conserved, strain background may influence metabolic vulnerabilities. The authors should discuss this, or ideally, test the genetic and chemical collateral vulnerabilities in at least one non-lineage 4 strain (RifR/ RifS) strain.

Version 2:

Reviewer comments:

Reviewer #2

(Remarks to the Author)

The authors have responded adequately to the technical comments/ suggestions from initial review.

Reviewer #3

(Remarks to the Author)

The authors have addressed my concerns. I don't have other comments.

Decision Letter:

Our ref: NMICROBIOL-25051584B

20th March 2026

Dear Jeremy,

Thank you for submitting your revised manuscript "Transcription attenuation amplifies collateral vulnerabilities in rifampicin-resistant *Mycobacterium tuberculosis*" (NMICROBIOL-25051584B). It has now been seen by the original referees and their comments are below. The reviewers find that the paper has improved in revision, and therefore we'll be happy in principle to publish it in Nature Microbiology, pending minor revisions to comply with our editorial and formatting guidelines.

Thank you again for your interest in Nature Microbiology. Please do not hesitate to contact me if you have any questions.

Reviewer #2 (Remarks to the Author):

The authors have responded adequately to the technical comments/ suggestions from initial review.

Reviewer #3 (Remarks to the Author):

The authors have addressed my concerns. I don't have other comments.

Version 3:

Decision Letter:

13th April 2026

Dear Jeremy,

I am pleased to accept your Article "Transcription attenuation amplifies collateral vulnerabilities in rifampicin-resistant *Mycobacterium tuberculosis*" for publication in Nature Microbiology. Thank you for having chosen to submit your work to us and many congratulations.

Authors may need to take specific actions to achieve compliance with funder and institutional open access mandates. If your research is supported by a funder that requires immediate open access (e.g. according to [a href="https://www.springernature.com/gp/open-science/plan-s-compliance"> Plan S principles](https://www.springernature.com/gp/open-science/plan-s-compliance) or the [a href="https://www.springernature.com/gp/open-science/us-federal-agency-compliance"> NIH public access policy](https://www.springernature.com/gp/open-science/us-federal-agency-compliance)) then you should select the gold OA route, and we will direct you to the compliant route where possible. Because authors warrant under our subscription licensing terms that they haven't committed to licensing any version of their article under a licence inconsistent with the terms of our agreement – including the applicable embargo period – publication under the subscription model isn't suitable for authors whose funders require no embargo.

An online order form for reprints of your paper is available at [a href="https://www.nature.com/reprints/author-reprints.html">https://www.nature.com/reprints/author-reprints.html](https://www.nature.com/reprints/author-reprints.html). All co-authors, authors' institutions and authors' funding agencies can order reprints using the form appropriate to their geographical region.

We welcome the submission of potential cover material (including a short caption of around 40 words) related to your manuscript; suggestions should be sent to Nature Microbiology as electronic files (the image should be 300 dpi at 210 x 297 mm in either TIFF or JPEG format). Please note that such pictures should be selected more for their aesthetic appeal than for their

scientific content, and that colour images work better than black and white or grayscale images. Please do not try to design a cover with the Nature Microbiology logo etc., and please do not submit composites of images related to your work. I am sure you will understand that we cannot make any promise as to whether any of your suggestions might be selected for the cover of the journal.

Congratulations once again and I look forward to seeing the article published.

P.S. Click on the following link if you would like to recommend Nature Microbiology to your librarian
<http://www.nature.com/subscriptions/recommend.html#forms>

** Visit the Springer Nature Editorial and Publishing website at http://editorial-jobs.springernature.com?utm_source=ejP_NMicro_email&utm_medium=ejP_NMicro_email&utm_campaign=ejp_NMicro for more information about our career opportunities. If you have any questions please click [here](mailto:editorial.publishing.jobs@springernature.com).

Reviewer #1:

This manuscript describes studies aimed at understanding collateral vulnerabilities and resistances arising in *M. tuberculosis* as a result of acquisition of rifampicin resistance. The authors have previously described these for the most common clinically acquired rifampicin allele (BS450L) and in this manuscript describe these for the next two most common alleles. At a high level it is a bit disappointing that there was little overlap between collateral changes amongst the three different changes in RNAP. The authors focused on understanding vulnerabilities unique to BS450L and elegantly show that these vulnerabilities are a result of enhanced termination of RNAP in a regulatory region controlling branch chain amino acid biosynthesis.

This is really solid and beautiful work, and I had only a few minor comments:

We thank the reviewer for their feedback.

(1) I wanted to dig in and see the full list of the highest differentially vulnerable amongst the three mutants (the blue dots in Fig 2) but the only place this is shown is in the source data in Supplemental Table 1. A full multi-tab 4000 gene complex dataset is pretty serious overkill for this, maybe a short table could be added (or even a tab that ties back to Figure 2) as I am certain others will want easy access to that gene list.

We thank the reviewer for this suggestion. To improve accessibility, we have added a new tab in Supplemental Table 1 that explicitly ties back to Figure 2, making it easier for readers to access these data.

(2) In the discussion the authors may want to point out that these allele-specific collateral changes may not be applicable to all kinds of targets, hopefully some metabolic targets will have collateral sensitivities that are more conducive to engineering combinations that could suppress emergence of resistance.

We agree this is an excellent point—that RifR may represent a unique case with respect to allele-specific collateral vulnerabilities. We have added a new paragraph to the Discussion to highlight this issue and to consider the possibility that other drug targets may yield more consistent collateral vulnerabilities that could be exploited to design resistance-suppressing combination therapies.

Reviewer #2:

Drug-resistance, and in particular Rifampin resistance in *Mtb* is a major global health threat. However, work in cancer as well as infectious diseases has identified that drug-resistant cells can be more susceptible to other agents – i.e. collateral vulnerability. In this work, Eckartt and colleagues build on the recent work from the Rock group that identified collateral vulnerabilities of the most common RifR mutation, RpoB-S450L. Here, they perform a similar experiment as performed previously for that strain, but with two other RifR mutant strains – in which the mutations cause faster as opposed to slower RNAP in an H37Rv background. Intriguingly, there are no (or v. few – it's unclear) shared collateral vulnerabilities between all 3 strains. It's not even clear if the two 'fast RNAP' strains share any vulnerabilities. Following up on S3450L vulnerabilities, the authors focus on branched-chain aa synthesis genes, esp *ilBb1*. Genetic or chemical inhibition of this gene leads to greater growth attenuation in the S450L strain compared with its WT parent. Further focusing the mechanism of this vulnerability, they confirm that it seems that *ilvB1* and its operon is regulated similarly to other model bacteria – i.e. there is a regulatory region upstream of the ORF that codes for a sORF that has a short stretch enriched for BCAA.

Overall, this is an elegant and well-performed study that confirms and follows up on a hit from the previous work of the group and shows that this vulnerability is 'unique' to this specific RifR mutation.

We thank the reviewer for their feedback.

Comments

1. Fig. 1: it seems that the H445Y strain has a very few vulnerabilities. I know that the data are in the supplement, but it's not readily apparent if any of the hits are shared with D435V. Can the authors call out the few hits (in blue) and explicitly clarify this point?

We thank the reviewer for this suggestion. The reviewer is correct: we identify very few collateral vulnerabilities in H445Y. There are no strong collateral vulnerabilities ($|\Delta V| \geq 3$) shared between H445Y and D435V. The only statistically significant collateral vulnerability shared between H445Y and D435V is ribonuclease E (*rne*; *rv2444c*), which is a weak hit in both fast RifR mutants (H445Y $\Delta V = -1.53$; D435V $\Delta V = -2.94$). Interestingly, *rne* is a modest but statistically significant collateral invulnerability in S450L ($\Delta V = 1.13$). Thus, *rne* knockdown produces a modestly higher fitness cost in H445Y and D435V, and a modestly lower fitness cost in S450L, relative to RifS Mtb. Ribonuclease E is an endoribonuclease that plays a central role in RNA processing and decay. The mechanistic basis for its divergent effects across these mutants remains unclear.

We have clarified this point in the revised Results section of the manuscript. We have also updated Supplemental Table 1 (also in response to Reviewer 1 comment 1) to make these data more accessible and highlight shared vs unique hits across all screened Mtb strains.

2. The lack of potential or major overlap between the two 'fast RNAP' strains is surprising. While it didn't become the major focus of this work, can the authors speculate more about this in the Discussion?

We have expanded our discussion of this phenomenon in the revised Discussion section.

3. And just to confirm, where there any shared hits between all three RifR strains?

Based on the cutoffs we apply in Figure 1C,D ($|\Delta V| \geq 3$), there are no shared collateral vulnerabilities or shared collateral invulnerabilities shared across all three RifR strains.

4. Fig. 4E: this experiment should be performed with the two other RifR strains, not just S450L, to confirm that S450L is the only one that is more sensitive to this drug.

We thank the reviewer for this suggestion. We have repeated the chlorflavonin MIC assays with the requested strains. As predicted by the genetic screen, β S450L was more sensitive to chlorflavonin than RifS, β D435V, or β H445Y Mtb. These data are now included in a revised Figure 4E and also shown below.

5. The proposed mechanism involves translation of a BCAA-rich region in cryptic uORF in the 5' UTR of the operon. If this is the case, one might predict that S450L RifR strains are more susceptible to leuRS inhibition also (which are commercially available). Although the mutation of the RifR is not stated, in this publication (PMID: 27503647), that did not seem to be the case. Could the authors test this agent or another LeuRS inhibitor with the 3 RifR strains and WT parent?

To address the reviewer's comment, we performed MIC assays with the LeuRS inhibitor Ganfeborole (GSK3036656). Consistent with the cited publication (PMID: 27503647), none of our tested RifR strains showed a significant change in MIC to GSK3036656. These data are shown in Response to Reviewer Figure 1 below.

6. Although the genetic architecture of this operon is conserved, strain background may influence metabolic vulnerabilities. The authors should discuss this, or ideally, test the genetic and chemical collateral vulnerabilities in at least one non-lineage 4 strain (RifR/ RifS) strain.

We appreciate the reviewer's point that strain background could influence metabolic vulnerabilities. Our current study was conducted in the reference lineage 4 strain H37Rv. In previously published work (PMID: 34297925), we found that *ilvB1* is also essential in the Lineage 2 clinical isolate HN878 and exhibits similar vulnerability (i.e. expression–fitness relationship) to that observed in H37Rv. Both strains were RifS in that analysis.

Based on our proposed mechanism, we do not expect the *ilvB1* collateral vulnerability phenotype to differ substantially across lineages. Our data indicate that this phenotype stems from altered RNA polymerase (RNAP) kinetics—specifically slower elongation and increased pausing and termination—caused by the *rpoB* S450L mutation. These effects on RNAP dynamics are almost certainly conserved among Mtb lineages; for instance, the core RNAP subunits (RpoA, RpoB, RpoC, and RpoZ) are 100% identical between H37Rv and HN878. Supporting this view, compensatory mutations in *rpoC*, *rpoA*, *nusG* and the *rpoB* β -protrusion domain—which mitigate the excessive pausing and termination associated with S450L—have arisen independently in multiple lineages (PMID: 38509362, 29661864). Together, these observations suggest that the *ilvB1* collateral vulnerability phenotype reflects a general consequence of altered RNAP dynamics rather than a lineage-specific phenomenon.

That said, to directly address the reviewer's suggestion and validate our findings in a non–lineage 4 background, we isolated spontaneous RifR mutants in the lineage 2 Mtb strain, HN878. From this, we recovered both *rpoB* β S450L and β H445Y mutants and confirmed by whole-genome sequencing that these were the only mutations present. MIC assays with chlorflavinin showed that, as in H37Rv, the β S450L mutant in HN878 exhibits increased sensitivity to *ilvB1* inhibition, whereas β H445Y does not. These data are now included as Supplemental Figure 3B in the revised manuscript and also shown below.

Reviewer #3:

This manuscript investigates the mechanism underlying increased gene vulnerability associated with the most prevalent rifampicin-resistance (Rif-R) mutation, β S450L, in *Mycobacterium tuberculosis*. The authors employed a comparative functional genomics approach to assess gene vulnerability across β S450L and two additional Rif-R *M. tuberculosis* strains. They identified the thiamine and branched-chain amino acid (BCAA) biosynthesis pathways as being uniquely susceptible in the β S450L mutant. Subsequently, they demonstrated that the β S450L mutation leads to transcriptional attenuation, which compromises the upregulation of *ilvB1* in response to both genetic and chemical inhibition of the BCAA biosynthetic pathway.

Overall, the manuscript is well written, and the findings are interesting. Although similar transcriptional regulation mechanisms have been described in other bacterial species, this is, to the best of my knowledge, the first report of such a mechanism in *M. tuberculosis*. This study might provide potential avenues for targeted intervention of β S450L RifR *M. tuberculosis* infection.

We thank the reviewer for their feedback.

Major comments:

1. Line 154-159. The authors categorize genes into two groups based on fitness differences between Rif-R and Rif-S *M. tuberculosis*: collateral vulnerability genes, which confer a greater fitness cost, and collateral invulnerability genes, which confer a smaller fitness cost. However, in my opinion, a more comprehensive classification should include three categories: (i) genes that impose a greater fitness cost, (ii) genes that confer a fitness advantage, and (iii) genes with minimal fitness impact. For example, in the β D435V strain, approximately 210 genes (< -1) are associated with increased fitness cost, while about 240 genes (> 1) appear to confer a growth advantage, with the remainder showing little change (Supplementary Table S1). In contrast, the β H445Y strain has only 27 genes associated with increased fitness cost, yet over 1,200 genes show a fitness advantage (> 1). This raises a concern: Why do so many genes appear to provide a growth advantage in β H445Y? Is this observation biologically meaningful, or might it reflect a limitation or bias in the method used to calculate the differences of gene vulnerability? Further clarification or validation is needed.

We appreciate the reviewer's comment and apologize for any confusion. To clarify, there are no genes for which knockdown confers an actual fitness advantage in either the RifS or RifR strains. All fitness differences (i.e. differential vulnerabilities) are measured relative to the control strain, not in absolute terms. Thus, a

positive differential fitness score does not indicate that gene knockdown improves growth; rather, it reflects that the magnitude of the fitness defect is smaller in the Rif^R background than in Rif^S.

For example, if a gene is essential in Rif^S, and its knockdown produces a greater fitness cost in β D435V, this gene is classified as a collateral vulnerability in β D435V, yet the gene remains essential in both strains. Conversely, if knockdown of an essential gene produces a smaller fitness cost in β D435V than in Rif^S, it is classified as a collateral invulnerability, but again, the gene is essential in both strains.

Under this framework, the large number of genes with positive differential fitness scores in β H445Y does not reflect fitness advantages but instead indicates that β H445Y experiences less severe fitness defects for many knockdowns relative to Rif^S. As the reviewer notes, β H445Y is indeed enriched for collateral invulnerabilities—though the extent of this skew appears overstated in their analysis, likely because a statistical significance cutoff (`sig_delta_vulnerability_index = True`) was not applied. We have now expanded our discussion of this pattern in both the revised Results and Discussion sections.

2. The BCAA pathway may be less susceptible in the β S450L strain that has acquired a compensatory mutation. Consequently, the effectiveness of targeted interventions could be limited in β S450L *M. tuberculosis* strains carrying such mutations. Additionally, the observed phenotype might differ in vivo. For instance, mutation of *rv0503c* was shown to increase rifampicin sensitivity in vitro (PMID: 32753506), but reduce sensitivity in vivo. Therefore, it would be advisable to validate these findings in cellular or animal models. At the very least, the authors should discuss it.

The reviewer is correct that the compensatory mutation we examined does reduce the sensitivity of the β S450L strain to the *ilvB1* inhibitor chlorflavonin. These data are presented in Figure 5H.

We also agree with the reviewer's important point that phenotypes observed in vitro may not fully reflect those in vivo. Due to biosafety restrictions, we are not permitted to perform in vivo experiments with drug-resistant *Mtb*. However, we explicitly discuss the relevance of targeting this pathway during infection in the third-to-last paragraph of the Discussion.

3. It would be helpful to express *ilvB1* under the control of a constitutive promoter to determine whether transcription attenuation alone accounts for the collateral vulnerability of *ilvB1* in the β S450L strain.

This is an excellent suggestion, and we devoted substantial effort to addressing it. We pursued three complementary approaches. First, we attempted to remove the Rho-independent terminator at the endogenous *ilvB1* locus in β S450L using single-strand DNA recombineering. Despite more than six months of effort, we were unable to obtain isolates suitable for the necessary downstream experiments.

As a second approach, we cloned the *ilvB1* operon under the control of its native promoter and 5' regulatory region (control; "WT") or generated a construct in which the sequence encoding the Rho-independent terminator (RIT) was deleted while preserving the antiterminator ("RIT-less"). We then tested whether removal of the RIT rescues the *ilvB1* collateral vulnerability phenotype in β S450L and found that it does. Together with the prior data presented in the manuscript, these results provide strong evidence that transcription attenuation contributes to the *ilvB1* collateral vulnerability observed in β S450L. These new data are included as Figure 5I in the revised manuscript and also shown below.

In parallel, we pursued an independent strategy to test the generality of the β S450L attenuation model by examining *whiB7*, the best-characterized example of transcription attenuation in Mtb (PMID: 16103351, 34687261, 38266648). Like *ilvB1*, *whiB7* expression is regulated by uORF-mediated transcription attenuation within its 5' regulatory region. Under non-stress conditions, transcription initiates at a distal upstream start site and proceeds through a short uORF. Efficient translation of this uORF promotes formation of a Rho-independent terminator, resulting in transcription termination upstream of the *whiB7* coding sequence. In contrast, during translation stress—such as exposure to macrolide antibiotics—translation stalling within the uORF favors formation of an antiterminator structure, allowing transcriptional readthrough into the *whiB7* ORF. Elevated WhiB7 levels then engage a positive feedback loop by activating transcription from the *whiB7* promoter and inducing the WhiB7 regulon.

To assess whether attenuation defects in β S450L extend beyond *ilvB1*, we examined *whiB7*. Consistent with our model, and similar to *ilvB1*, β S450L exhibits impaired upregulation of *whiB7*. Notably, although *whiB7* induction is reduced in β S450L, this defect does not result in pronounced hypersensitivity to translation inhibitors under standard MIC conditions. Nevertheless, just as the thiamine-related collateral vulnerability can be explained, at least in part, by diminished *ilvB1* induction in β S450L, the collateral vulnerability associated with *aspC*—a direct WhiB7-activated gene—may similarly reflect impaired *whiB7* activation in β S450L. Together, these findings indicate that attenuation defects in β S450L are generalizable and can propagate through regulatory circuits and metabolically connected pathways, producing cascading effects on downstream genes. These data are now included as a new Supplemental Figure 6 and are discussed in the revised Discussion section.

Minor comment:

1. In Figure 2A, the y-axis label should be corrected to “V_RifR-V_RifS”.

We thank the reviewer for catching this mistake. This has now been corrected.

2. Differential vulnerability results of H445Y/D435V and S450L should also be presented in supplementary table.

We thank the reviewer for this suggestion. We have now included differential vulnerability results of H445Y/D435V and S450L in a revised Supplemental Table 1.